https://doi.org/10.1038/s43856-023-00270-4　　OPEN
# Fusion peptide is superior to co-expressing subunits for arming oncolytic herpes virus with interleukin 12

Hiroshi Fukuhara [1,2], Yuzuri Tsurumaki Sato [1], Jiangang Hou [1], Miwako Iwai [1] & Tomoki Todo [1✉]

## Abstract

**Background** G47Δ is a triple-mutated oncolytic herpes simplex virus type 1 (HSV-1) recently approved as a new drug for malignant glioma in Japan. As the next-generation, we develop armed oncolytic HSV-1 using G47Δ as the backbone. Because oncolytic HSV-1 elicits specific antitumor immunity, interleukin 12 (IL-12) can function as an effective payload to enhance the efficacy.

**Methods** We evaluate the optimal methods for expressing IL-12 as a payload for G47Δ-based oncolytic HSV-1. Two new armed viruses are generated for evaluation by employing different methods to express IL-12: T-mfIL12 expresses murine IL-12 as a fusion peptide, with the genes of two subunits (p35 and p40) linked by bovine elastin motifs, and T-mIL12-IRES co-expresses the subunits, with the two genes separated by an internal ribosome entry site (IRES) sequence.

**Results** T-mfIL12 is significantly more efficient in producing IL-12 than T-mIL12-IRES in all cell lines tested, whereas the expression methods do not affect the replication capabilities and cytopathic effects. In two syngeneic mouse subcutaneous tumor models of Neuro2a and TRAMP-C2, T-mfIL12 exhibits a significantly higher efficacy than T-mIL12-IRES when inoculated intratumorally. Furthermore, T-mfIL12 shows a significantly higher intratumoral expression of functional IL-12, causing stronger stimulation of specific antitumor immune responses than T-mIL12-IRES.

**Conclusions** The results implicate that a fusion-type expression of IL-12 is a method superior to co-expression of separate subunits, due to higher production of functional IL-12 molecules. This study led to the creation of triple-mutated oncolytic HSV-1 armed with human IL-12 currently used in phase 1/2 trial for malignant melanoma.

## Plain language summary

Some viruses, including the herpes virus, can be modified so that they can target and kill cancers. These viruses can be loaded with factors that stimulate the immune system, which can help to eradicate cancer cells. Here, we test different methods of loading a cancer-killing version of the herpes virus with interleukin 12, an immune-stimulating factor. We show that one method, which involves loading the virus with the different parts of interleukin 12 fused together, is superior to another, and leads to improved anti-cancer effects in mouse models. These findings have contributed to the creation of a cancer-killing virus that is currently in clinical trials in patients with melanoma.

[1] Division of Innovative Cancer Therapy, Advanced Clinical Research Center, The Institute of Medical Science, The University of Tokyo, 4-6-1 Shirokanedai, Minato-ku, Tokyo 108-8639, Japan. [2] Department of Urology, Kyorin University School of Medicine, 6-20-2 Shinkawa, Mitaka, Tokyo 181-8611, Japan. ✉email: toudou-nsu@umin.ac.jp

Oncolytic herpes simplex virus type 1 (HSV-1) is an effective tool for treating cancers[1,2]. The first oncolytic HSV-1 with a deletion in the thymidine kinase gene was described in 1991[3]. Since then, various oncolytic HSV-1 have been tested in clinical trials. T-VEC (Talimogene laherparepvec) is an oncolytic HSV-1 with mutations in the γ34.5 and α47 genes, and was approved as a drug for inoperable melanoma in the United States and Europe in 2015[4]. G207, one of the first oncolytic HSV-1 viruses to be used in clinical trials, has deletions in both copies of the γ34.5 gene and a lacZ insertion inactivating the ICP6 gene[5]. G47Δ was constructed by creating a further deletion in the α47 gene and the overlapping US11 promoter of the G207 genome[6]. The efficacy of G47Δ in various cancers has been demonstrated[7–16]. The safety of G47Δ has been tested in several clinical trials in Japan, including those for glioblastoma (UMIN 000002661)[17], castration-resistant prostate cancer (UMIN0000104 63), metastatic prostate cancer (jRCTs033210603), olfactory neuroblastoma (UMIN000011636, jRCTs033180325), and malignant pleural mesothelioma (UMIN000034063, jRCTs033180326). Recently, the phase II trial (UMIN000015995) for glioblastoma led to the approval of G47Δ as a new drug for malignant glioma in Japan[18].

Interleukin-12 (IL-12) is a promising tool for cancer treatment, and several types of murine IL-12 has been used to be expressed by oncolytic HSV-1[19–22]. In general, tumor formation and growth depend largely on the host's inability to elicit a potent antitumor immune response, and on the formation of new blood vessels that nourish the tumor. IL-12 can target both processes. IL-12 is a cytokine that activates various immune cells, including NK and T cells. In effect, IFN-γ is induced and secreted by these cells. Intact IL-12, p70, is a heterodimeric glycoprotein consisting of two subunits, a 35 kDa light chain (p35) and a 40 kDa heavy chain (p40)[23,24]. The p70–IL-12Rβ1 complex binds to the second p70, and the resulting complex then presents a preformed binding site with high affinity for IL-12Rβ2. The interactions between p40 and IL-12Rβ1 is needed to stabilize the p70–IL-12Rβ1 complex[25]. A simultaneous transfection of mammalian cells with these two subunit genes is necessary for the production of biologically active IL-12[26,27]. In this study, we prepared two different murine IL-12 expression cassettes, both with an immediate-early cytomegalovirus (CMV) promoter: 1) The p35 and p40 subunit genes are linked as one gene in the same reading frame, resulting in an expression of a fusion peptide comprising p40 and p35: 2) The p35 and p40 subunit genes are separated by an internal ribosome entry site (IRES) sequence, resulting in co-expression of two separate proteins of p40 and p35 subunits. Since the p35 subunit does not appear to be secreted in the absence of the p40 subunit even though its transcripts are present in many cell types[28], the p40 gene was placed closer to the promoter. For the fusion protein of two subunits, whether the expressed molecule maintains a functional tertiary structure is a critical issue. It is devised for the two subunits to fold and interact by inserting two bovine elastin motifs between the p35 and p40 genes. IRES, on the other hand, ensures that two subunits expressed separately would form a molecule with a preserved tertiary structure and with an intact IL-12 function. However, IRES might cause an uneven expression of the two subunit genes, leading to an overexpression of the p40 gene closer to the promoter. An excess-free p40 subunit might inhibit the activity of IL-12 in murine models[29].

The α47 gene product binds to the transporter associated with antigen presentation (TAP) and down-regulates the expression of major histocompatibility complex (MHC) class I molecules, so as to avoid detection of the infected cell by host immune surveillance[30]. G47Δ, an α47-deficient HSV-1, prevents the down-regulation of MHC class I and exhibits an enhanced immune stimulation capability[6], and is therefore suited as the backbone to express immunostimulatory transgenes in humans. In this study, the two different murine IL-12 expression cassettes were each inserted into the deleted ICP6 locus of the G47Δ backbone to create two types of IL-12-expressing oncolytic HSV-1, T-mfIL12 and T-mIL12-IRES. The difference in efficacy should reflect the methods we employed to express murine IL-12. When evaluated in vitro and in vivo using murine prostate cancer and neuroblastoma cells, T-mfIL12, a fusion-type IL-12, shows higher expression of functional IL-12 and higher antitumor activities than T-mIL12-IRES.

## Methods

**Cells.** TRAMP-C2 cells were obtained from the American Type Culture Collection (ATCC; Rockville, USA). The cells were grown in DMEM supplemented with 5% (v/v) fetal calf serum (FCS), 5% (v/v) Nu-serum IV and 10 nM dihydrotestosterone. Pr14-2 cells were a kind gift from Dr. Jeffery Green (NCI, Bethesda, USA) and were cultured in DMEM supplemented with 10% (v/v) FCS. Vero (African green monkey kidney) and Neuro2a cells were obtained from the ATCC and maintained in DMEM supplemented with 10% (v/v) FCS and 2 mM glutamine.

**Virus construction.** Conventionally, recombinant HSV-1 have been constructed using homologous recombination techniques, which required time and effort-consuming processes of selecting and confirming the structures of a desired recombinant that occurred at a very low probability, because of the large genome size of HSV-1. In order to eliminate the element of chance as much as possible, we previously constructed the G47Δ-BAC system using bacterial artificial chromosome (BAC) and two recombinases, allowing the development of multiple armed oncolytic HSV-1 viruses in parallel, swiftly and precisely, using G47Δ as the backbone[31]. However, G47Δ-BAC products were later found to have lower replication capabilities than G47Δ, presumably due to unknown mutations in the G47Δ-BAC plasmid. We therefore reconstructed the entire system under the same concept, but with added modifications, including useful multiple cloning sites. This T-BAC system allows the production of G47Δ-derived armed oncolytic HSV-1 with replication capabilities comparable to G47Δ and with desired transgenes. First, the parental T-BAC virus was created by homologous recombination of G47Δ DNA and pBAC-ICP6EF[31], a plasmid that contains the insertion sequences of the ICP6 coding region. Transfections were performed on Vero cells using 0.9 μg of DNA, composing a 1:1:1 mixture of G47Δ DNA purified by Na/I method, pBAC-ICP6EF (undigested), and pBAC-ICP6EF linearized with AscI digestion, with Lipofectamine (11668027, Thermo Fisher Scientific, USA), according to the manufacturer's instructions. At a 30–50% cytopathic effect, recombinant viruses forming GFP-positive plaques were selected and further passaged in Vero cells. After three rounds of GFP positive and lacZ negative selection, we selected the parental T-BAC virus by screening several candidates that replicated well. This step is important as the altered virus has unexpected mutations that reduce the virus yield. Circular viral DNA from infected Vero cells was isolated by the Hirt method and electroporated into E.coli DH10B (18290015, Thermo Fisher Scientific, USA). Antibiotic-resistant colonies were isolated, the T-BAC plasmid DNA was purified, and the structure was confirmed by endonuclease digestions.

Second, we prepared two plasmids to fill a shuttle vector (pVec9) with a murine IL-12 expression cassette. The fusion type of murine IL-12 gene in T-mfIL12 is derived from the pORF-mIL-12 plasmid (porf-mil12, InvivoGen, San Diego, USA) in which the p35 and p40 subunits are linked by bovine elastin motifs. The fusion-type

murine IL-12 is expressed as a single peptide with the signal sequence in the p35 subunit. We developed a pVec9-fused-mIL12 plasmid using a plasmid carrying the p35 and p40 genes of murine IL-12 as one gene in the same reading frame. A 1.7 kb PCR fragment from pORF-mIL-12 (primers: 5'-GCTAGCCTGAGATCACCGGCG-3', 5'-GCTAGCATCCGTTGCATCCTA-3') was digested with NheI and inserted into the AvrII site of pVec9 to generate pVec9-fused-mIL12. Similarly, we constructed a pVec9-IRES-mIL12 plasmid using a single expression cassette, separated by an IRES sequence from the 5' non-translated region of encephalomyocarditis virus (EMCV). The IRES type of murine IL-12 gene in T-mIL12-IRES was derived from the plasmid pHCIL12-tk[32]. The constructs were verified by sequencing. A 2.3 kb SpeI-StuI fragment containing the pIL-12-p40-IRES-p35 plasmid was inserted into the AvrII-StuI site of pVec9 to generate pVec9-IRES-mIL12.

Third, a mixture of the T-BAC plasmid (1.5 μg) and pVec9-fused-mIL12 or pVec9-IRES-mIL12 (150 ng each) was incubated with Cre recombinase (M0298, NEB, USA) at 37 °C for 30 min in 10 μl of the solution and electroporated into E. coli DH10B. To select those that contained the mutant BAC plasmid, the bacteria were streaked onto LB plates containing Cm (15 μg/ml) and Kan (10 μg/ml) and incubated at 37 °C overnight. DNA structures of the recombinant T-BAC/Vec9 plasmids were confirmed by gel analyses following endonuclease digestion. Transfections were performed on Vero cells by using 2 μg of T-BAC/Vec9 DNA and 0.5 μg of pOG44, the plasmid expressing Flp (V600520, Thermo Fisher Scientific, USA), with 15 μl of Lipofectamine, according to the manufacturer's instructions. Transfected cells were incubated in DMEM with 10% (v/v) FCS at 37 °C overnight, and the medium was replaced with DMEM containing 1% heat-inactivated fetal calf serum (IFCS) the following day. Incubation was continued for several days until plaques appeared. The progeny viruses were selected for GFP negativity under inverted fluorescence microscope and lacZ positivity by X-gal staining. Three rounds of limiting dilution were performed to pick out a single clone. Recombinant viruses were harvested, and the structure of the viral DNA was confirmed by endonuclease digestion. Transgene cassettes of the recombinant viruses were sequenced (primers: 5'-CGCAAATGGGCGGTAGGCGTG-3', 5'-TAGAAGGCACAGTCGAGG-3', 5'-ACCCGCCCAAGAACTTGCAG-3', 5'-GGATCGGACCCTGCAGGGAAC-3'). Primers were designed to sequence the nucleotides between the CMV promoter and polyA. These viruses were individually titrated on Vero cells by plaque assay.

**Replication assay and cytopathic effect studies**. For the replication assay, Vero cells were seeded on 6-well plates at a density of $3 \times 10^5$ cells per well. Wells were infected with G47Δ, T-01, four clones of T-mfIL12 or four clones of T-mIL12-IRES in duplicate wells at an MOI of 0.01. After 48 h of infection, the cells were lysed by three cycles of freezing and thawing. The progeny virus was titrated on Vero cells by plaque assay as described previously[6]. In a separate experiment, Vero cells were infected with G47Δ, T-01, T-mfIL12 or T-mIL12-IRES in duplicate wells at an MOI of 0.01, the progeny virus was recovered 0 h, 6 h, 24 h and 48 h after infection, and titrated by plaque assay. Results represent the arithmetic mean of experiments conducted in duplicate. Cytopathic effect study was performed as described previously[6]. Cells were seeded onto six-well plates and incubated at 37 °C overnight. The cells were then inoculated with the virus for 1 h. The inoculum was removed, and cells were incubated in DMEM. The number of surviving cells were counted daily with a Coulter Counter (Z1 single, Beckman Coulter, USA) and expressed as a percentage of mock-infected controls.

**In vitro IL-12 expression measurement**. Vero, Neuro2a, Pr14-2 or TRAMP-C2 cells were plated in 24-well plates ($1 \times 10^5$ cells/well) and incubated at 37 °C for 24 h. Cells in duplicate wells were infected with each virus at MOI = 1 and further incubated at 39.5 °C for 48 h. All viruses used in this study derive from HSV-1 strain F, are therefore temperature sensitive, and do not replicate at 39.5 °C[5]. Supernatants were collected, and the concentration of IL-12 was measured using mouse IL12 p70 immunoassay (M1270, R&D Systems Inc., IL) with a detection limit of 7.8 pg/ml. Results represent the arithmetic mean of experiments conducted in duplicate. For the detection of the p35 subunit and p40 subunit separately, ELISA kit for mouse IL12A (IL12 p35) (SEA059Mu, Cloud-Clone, TX) and ELISA kit for mouse IL12B (IL12 p40) (SEA058Mu, Cloud-Clone, TX) were used. The expression ratios of p35 and p40 subunits were adjusted based on the control (recombinant mouse IL-12, 095-05331, Wako, Japan).

**Interferon-γ release assay**. Supernatants of Vero cells infected with T-mfIL12 or T-mIL12-IRES at MOI = 1 were collected 48 h post-infection. VP-SFM medium (11681020, Thermo Fisher Scientific) was used for dilution. The spleen from A/J mice was harvested, and cell suspensions were prepared. Splenocytes were subjected to the supernatants or recombinant mouse IL-12 (rIL12, 095-05331, Wako, Japan) for 48 h and interferon-γ (IFNγ) levels were measured in duplicates using mouse IFNγ Uncoated ELISA kit (88-7314, ThermoFisher Scientific).

**Animal experiments**. Six-week-old male C57BL/6 mice, female A/J mice and female NOG (NOD/shi-scid.IL-2RγKO) mice were purchased from CLEA Japan (Tokyo, Japan). All animals were caged in groups of five or less. Subcutaneous tumor therapy was performed as described previously[31]. Subcutaneous tumors were generated by implanting TRAMP-C2 prostate cancer cells ($5 \times 10^6$) into the left flank of male C57BL/6 mice or Neuro2a neuroblastoma cells ($5 \times 10^6$) into bilateral flanks of female A/J mice or NOG mice. When the tumors reached approximately 5 mm in diameter, 5–7 days after implantation, animals were randomly divided into groups and blinded, and mock, virus suspension or recombinant mouse IL-12 p70 (#577004, BioLegend; rIL-12) in 20 μl was inoculated into the left tumors. Treatments were repeated 3 or 4 days later. Tumor growth was determined by measuring tumor volume (length × width × height × 0.52) twice a week. Animals were sacrificed when the maximum diameter of the tumor reached 22 mm. All procedures involving animals were approved by the Institutional Animal Care and Use Committee of Graduate School of Medicine and Faculty of Medicine, The University of Tokyo (approval number, P12-84).

**Immunohistochemistry**. Bilateral subcutaneous Neuro2a tumors were generated in A/J mice, left tumors only were inoculated with T-mfIL12, T-mIL12-IRES, T-01 ($2 \times 10^5$ pfu) or mock on days 0 and 3, and the tumors were harvested on day 6 (n = 3 per group). Tumors were fixed in 20% formaldehyde overnight, embedded in paraffin and sectioned at 4 μm. Sections were immunostained with anti-CD8 antibody (98941, CST, Massachusetts, USA), anti-CD4 antibody (25229, CST, Massachusetts, USA) or anti-HSV-1 antibody (ab9533, Abcam, Cambridge, United Kingdom) and developed with diaminobenzidine.

**In vivo IL-12 and IFNγ levels**. In the unilateral subcutaneous Neuro2a model, T-01, T-mfIL12, T-mIL12-IRES ($2 \times 10^6$ pfu) or mock was inoculated into established tumors on day 0. Sera and tumor samples were collected from three mice per group on days 1, 3 and 6, and the mouse IL-12 and mouse IFNγ levels were

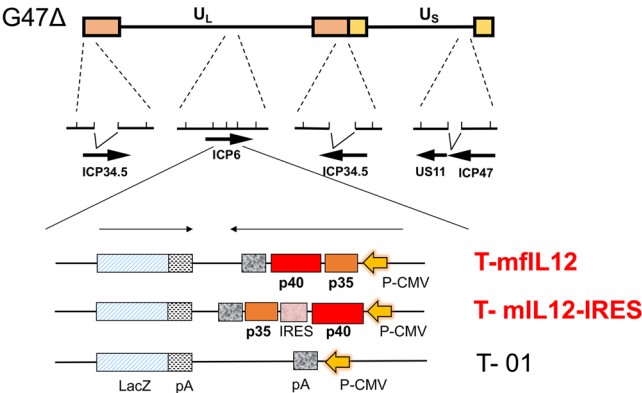

**Fig. 1 The structures of T-mfIL12 and T-mIL12-IRES.** The boxes (top line) represent inverted repeat sequences flanking the long (U$_L$) and short (U$_S$) unique sequences of HSV-1 DNA. Oncolytic HSV-1 uses G47Δ as the backbone and contains 1.0 kb deletions in both copies of the γ34.5 gene, a 0.3 kb deletion in the α47 gene, and a 0.9 kb deletion in the ICP6 gene. The lacZ gene and the cytomegalovirus (CMV) promoter-driven interleukin-12 (IL-12) gene are oriented in opposite directions, and inserted into the deleted ICP6 locus. T-mfIL12 contains a cassette carrying the p35 and p40 genes of murine IL-12 linked by two bovine elastin motifs. Murine IL-12 is expressed as a single fusion peptide and folds to become active. T-mIL12-IRES contains a cassette in which the p40 and p35 genes of murine IL-12 are separated by an internal ribosome entry site (IRES) sequence from the encephalomyocarditis virus (EMCV). The p40 and p35 subunits are efficiently co-expressed to form active murine IL-12. T-01 contains a cassette with no transgene.

measured in duplicate using mouse IL12 p70 immunoassay (M1270, R&D Systems Inc., IL) and mouse IFNγ Uncoated ELISA (88-7314, ThermoFisher Scientific). Tumor samples were also collected for T-mfIL12 and T-mIL12-IRES groups on day 0, immediately after virus injections ($n = 4$). The base IL-12 concentrations in mock and virus suspensions were measured by ELISA: IL-12 was not detected in mock and T-01, and the amounts of IL-12 in 20 µl ($2 \times 10^6$ pfu) of T-mfIL12 and T-mIL12-IRES were calculated to be 157.3 pg and 5.2 pg, respectively.

**Enzyme-linked immunospot (ELISpot) assay.** In the unilateral subcutaneous Neuro2a model, T-01, T-mfIL12, T-mIL12-IRES ($5 \times 10^4$ pfu) or mock was injected intratumorally on days 0 and 3. On day 6, the spleen was harvested aseptically from three mice per group and cell suspensions were prepared. Splenocytes were cocultured with Neuro2a cells or A/J mouse-derived Sal/N cells (negative control) and subjected to ELISpot assay using Mouse Interferon-γ ELISpot$^{PLUS}$ (3321-4HPW-2, Mabtech AB, Sweden) and Mouse interleukin-4 (mIL-4) ELISpot$^{PLUS}$ (3311-4HPW-2, Mabtech AB, Sweden). Assay was performed in triplicates.

**Statistics and reproducibility.** Data comparisons between treatment groups were made using Student's t-test, one-way ANOVA with Tukey's multiple comparisons test or two-way ANOVA with Bonferroni's multiple comparisons test. Survival studies were analyzed by Kaplan-Meier analysis. A value of $P < 0.05$ was considered statistically significant. All statistical analyses were performed using JMP Pro version 11.0.0 (SAS Institute, USA). The experiments were conducted in duplicate for replication assay, cytopathic effect studies, and in vitro expression measurements. Tumor samples were collected from three mice per group for the measurements of in vivo IL-12 and IFNγ levels. For in vivo animal experiments, nine to twelve mice were used per

group. Three mice per group were treated for IFNγ and IL-4 ELISpot assays, and assays were performed in triplicates.

**Reporting Summary**. Further information on research design is available in the Nature Portfolio Reporting Summary linked to this article.

## Results

**Construction of T-mfIL12 and T-mIL12-IRES.** We previously utilized a bacterial artificial chromosome (BAC) and two recombinase systems (Cre/loxP and Flp/FRT) to construct oncolytic HSV-1 armed with murine interleukin 18 and soluble murine B7-1 using G47Δ as the backbone[31]. The system was since reconstructed with modifications (T-BAC system) so as to produce G47Δ-derived armed oncolytic HSV-1 with improved replication capabilities and with desired transgenes. Using this T-BAC system, we generated two types of IL-12-expressing oncolytic HSV-1, T-mfIL12 and T-mIL12-IRES, utilizing two different murine IL-12 expression cassettes. Each virus has an 894 bp deletion in the ICP6 gene where the expression cassette is inserted, in addition to deletions in both copies of the γ34.5 gene and a deletion in the α47 gene (Fig. 1). The lacZ gene is placed behind the ICP6 promoter and the expression cassette controlled by the CMV promoter is placed in the opposite direction. The size of the murine IL-12 gene in T-mfIL12 from the ATG to the stop codon is 1,620 bp, and the p35 gene in the upstream is linked with the p40 gene at the downstream, so the murine IL-12 is expressed as a fusion protein. The murine IL-12 gene in T-mIL12-IRES contains 1,008 bp of p40 subunit in front and 648 bp of p35 subunit at the back with the IRES sequence in between, so the two subunits are expressed simultaneously but separately. T-01, a control virus, was constructed using the shuttle vector with an empty cassette. These three viruses were constructed in parallel, and more than 99% of viral plaques formed after recombination were positive for a lacZ expression. Four clones of each recombinant HSV-1 were isolated by three rounds of limiting dilution, and the construct was confirmed by restriction endonuclease digestion (Supplementary Fig. 1). Transgene cassettes of T-mfIL12 and T-mIL12-IRES were sequenced to confirm that the sequences of the p35 subunit and the p40 subunit of T-mfIL12 were identical to those of T-mIL12-IRES (Supplementary Note 1).

**In vitro characteristics of T-mfIL12 and T-mIL12-IRES.** To assess the replication capability of the two types of viruses expressing IL-12, we measured the number of progeny virus recovered from Vero cells ($3 \times 10^5$ cells per well) 48 h after infection at an MOI of 0.01 (Fig. 2a, Supplementary Data 1). The arithmetic mean of virus yields (±SD) obtained from 4 clones of T-mfIL12 and 4 clones of T-mIL12-IRES were $6.8 \times 10^6$ (±$0.59 \times 10^6$) and $7.0 \times 10^6$ (±$0.94 \times 10^6$) pfu, respectively ($p = 0.736$, t-test). These yields were comparable to that of the control virus T-01 ($7.6 \times 10^6$ pfu), whereas the parental G47Δ showed a higher yield ($2.0 \times 10^7$ pfu). The amount of murine IL-12 expression was assessed by measuring the concentration in the supernatants by ELISA 48 h after infecting Vero cells at an MOI of 1 (Fig. 2b, Supplementary Data 1). The ELISA specifically detects the functional form of murine IL-12 (p70). The arithmetic mean of IL-12 concentrations (±SD) of 4 clones of T-mfIL12 and 4 clones of T-mIL12-IRES were $10.0 \pm 0.18$ and $1.5 \pm 0.39$ ng/mL, respectively ($p < 0.001$, t-test), T-mfIL12 expressing the p70 form of IL-12 almost 7-fold higher than T-mIL12-IRES. The first clone identified from each recombinant G47Δ was used for further analysis. The adjusted expression ratios of p35 and p40 subunits were 1.8:1 and 10:1 for T-mfIL12 and T-mIL12-IRES, respectively. The time course (0 h, 6 h, 24 h and 48 h) for the viral yields

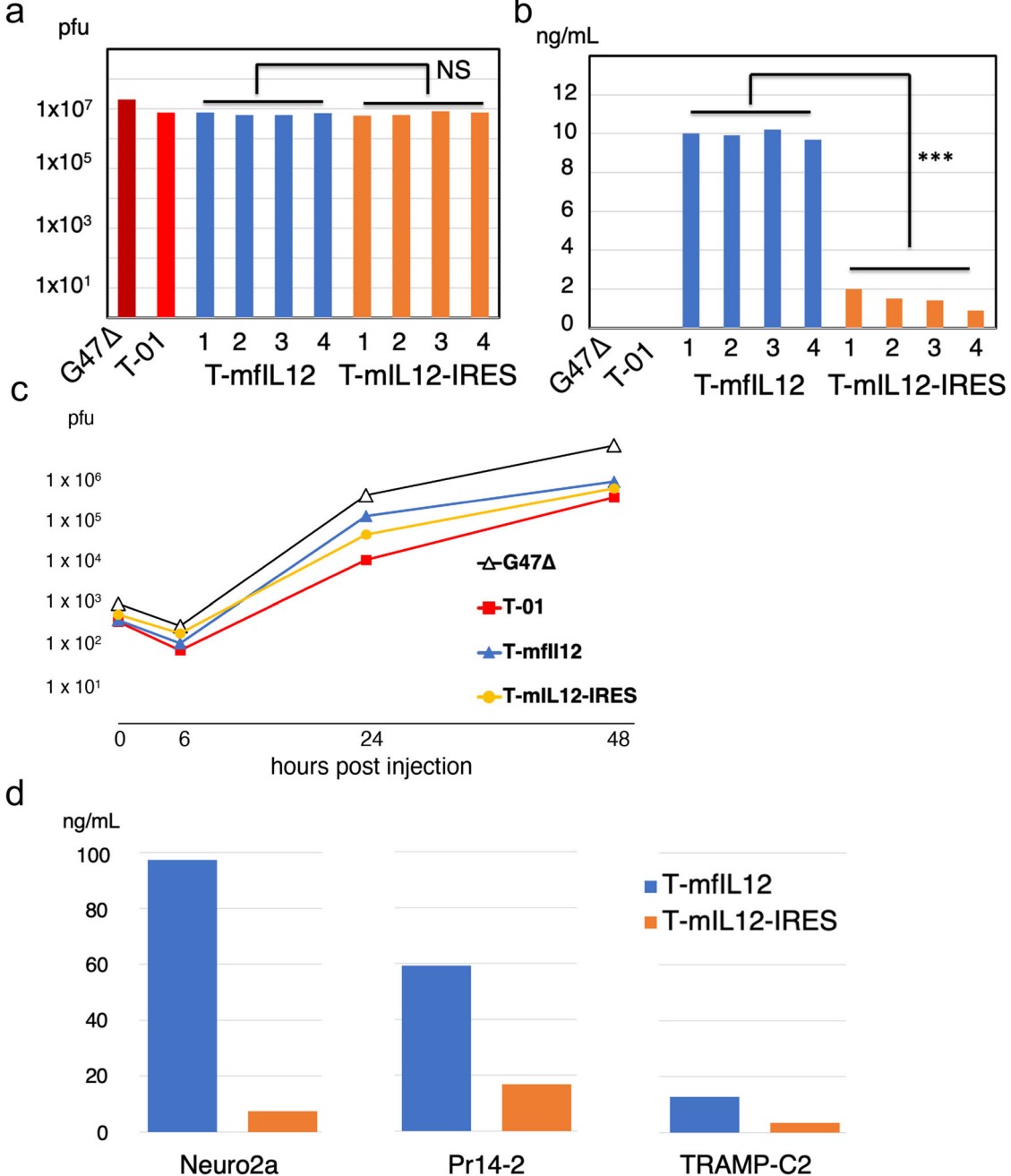

**Fig. 2 In vitro replication capabilities and murine IL-12 expressions of T-mfIL12 and T-mIL12-IRES. a** In vitro replication assay. Vero cells were infected with G47Δ, T-01, T-mfIL12 candidate (4 clones) or T-mIL12-IRES candidate (4 clones) at a multiplicity of infection (MOI) of 0.01, the progeny virus was recovered 48 h after infection, and the number determined by plaque assay. T-mfIL12 and T-mfIL12-IRES showed no significant difference in replication capability (p = 0.736, t-test). **b** In vitro murine IL-12 expression. Vero cells were infected with G47Δ, T-01, T-mfIL12 candidate (4 clones) or T-mIL12-IRES candidate (4 clones) at an MOI of 1, and the amount of murine interleukin-12 (IL-12) (p70) secreted was determined by enzyme-linked immune-sorbent assay (ELISA). T-mfIL12 expressed a significantly higher amount of p70 IL-12 than T-mIL12-IRES (p < 0.001, t-test). **c** The time course of viral yields. Vero cells were infected with G47Δ, T-01, T-mfIL12 or T-mIL12-IRES in duplicate at an MOI of 0.01, the progeny virus was recovered 0 h, 6 h, 24 h and 48 h after infection, and titrated by plaque assay. The time course for the viral yields of T-mfIL12 was comparable to that of T-mIL12-IRES. **d** In vitro murine IL-12 expression in murine tumor cell lines. Neuro2a, Pr14-2 or TRAMP-C2 cells were infected with T-mfIL12 or T-mIL12-IRES at an MOI of 1, and the amount of murine IL-12 (p70) secreted was determined by ELISA. T-mfIL12 expressed a higher amount of p70 IL-12 than T-mIL12-IRES in all three cell lines. All assays were performed in duplicate. ***, p < 0.001; NS, not significant.

of T-mfIL12 in Vero cells was comparable to that of T-mIL12-IRES (Fig. 2c, Supplementary Data 1). IL-12 expressed by T-mfIL12 and T-mIL12-IRES both stimulated splenocytes, causing a release of IFNγ, and were thus confirmed bioactive (Supplementary Fig. 2, Supplementary Data 2).

The in vitro cytopathic effects of T-mfIL12 and T-mIL12-IRES were assessed using a mouse neuroblastoma cell line Neuro2a and mouse prostate cancer cell lines Pr14-2 and TRAMP-C2. In all three murine cancer cell lines, both T-mfIL12 and T-mIL12-IRES showed cytopathic effects comparable to the control virus T-01 at

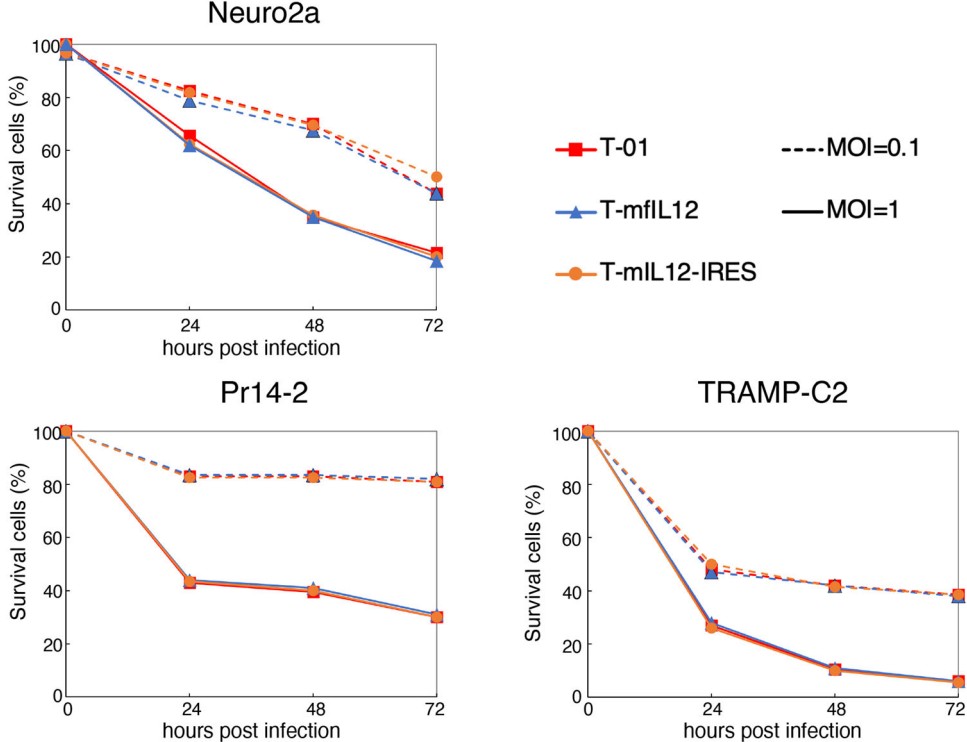

**Fig. 3 In vitro cytopathic effects of T-mfIL12 and T-mIL12-IRES.** In all three murine cancer cell lines, both T-mfIL12 and T-mIL12-IRES showed cytopathic effects comparable to the control virus T-01 at both multiplicity of infection (MOI) = 0.1 (dotted lines) and MOI = 1 (solid lines). The number of surviving cells was counted daily and expressed as a percentage of mock-infected controls. The results are the means of triplicate wells.

both MOIs 0.1 and 1 (Fig. 3, Supplementary Data 3). Expressions of murine IL-12 by T-mfIL12 and T-mIL12-IRES were further evaluated in the three cell lines. The concentrations in the supernatants were measured by ELISA 48 h after infection at an MOI of 1 (Fig. 2d, Supplementary Data 1). In all three cell lines, T-mfIL12 showed expressions of murine IL-12 higher than T-mIL12-IRES.

**In vivo efficacy of T-mfIL12 and T-mIL12-IRES.** The in vivo efficacy of the two types of IL-12-expressing viruses was evaluated in two immunocompetent mouse tumor models, TRAMP-C2 tumors in syngeneic C57BL/6 mice and Neuro2a tumors in syngeneic A/J mice (Fig. 4a, b, Supplementary Fig. 3, Supplementary Data 4). When established subcutaneous TRAMP-C2 tumors reached approximately 5 mm in diameter, T-01, T-mfIL12, T-mIL12-IRES ($5 \times 10^6$ pfu) or mock were inoculated intratumorally twice on days 0 and 3. C57BL/6 mice and C57BL/6-derived TRAMP-C2 cells are relatively resistant to HSV-1 infection[33], and the dose ($5 \times 10^6$ pfu) was determined based on our previous studies[9]. In this TRAMP-C2 model, the T-mfIL12 treatment showed the highest efficacy, resulting in a significantly smaller tumor than T-01 and T-mIL12-IRES treatments on day 25 ($p = 0.003$ and $p = 0.029$ versus T-01 and T-mIL12-IRES, respectively, t-test; Fig. 4a). Between groups, all three viruses were significantly more efficacious than mock ($p < 0.001$ for all three viruses, two-way ANOVA), however only T-mfIL12, but not T-mIL12-IRES, was significantly more efficacious than T-01 ($p = 0.027$, two-way ANOVA; Fig. 4a).

Poorly immunogenic Neuro2a cells were implanted in the bilateral flanks of A/J mice. When established subcutaneous Neuro2a tumors reached approximately 5 mm in diameter, T-01, T-mfIL12, T-mIL12-IRES ($5 \times 10^4$ pfu) or mock was inoculated into the left tumors only, twice on days 0 and 3. A/J mice and A/J-

derived Neuro2a cells are susceptible to HSV-1 infection[33], and the dose ($5 \times 10^4$ pfu) was determined based on our experience with this model[20]. Both T-mfIL12 and T-mIL12-IRES caused a significant tumor growth inhibition compared with T-01 in the treated side on day 11 ($p < 0.001$ and $p = 0.016$ for T-mfIL12 and T-mIL12-IRES, respectively, t-test) as well as in the untreated side on day 11 ($p < 0.001$ and $p = 0.018$ for T-mfIL12 and T-mIL12-IRES, respectively, t-test; Fig. 4b). Between groups, all three viruses were significantly more efficacious than mock in the treated side ($p = 0.001$ for T-01 and $p < 0.001$ for both T-mfIL12 and T-mIL12-IRES, two-way ANOVA) and T-mfIL12 and T-mIL12-IRES than mock in the untreated side ($p < 0.001$ for both viruses, two-way ANOVA; Fig. 4b). Also, both T-mfIL12 and T-mIL12-IRES were significantly more efficacious than T-01 in the treated side ($p < 0.001$ and $p = 0.006$ for T-mfIL12 and T-mIL12-IRES, respectively, two-way ANOVA) as well as in the untreated side ($p < 0.001$ and $p = 0.011$ for T-mfIL12 and T-mIL12-IRES, respectively, two-way ANOVA). Furthermore, T-mfIL12 caused a significant tumor growth inhibition compared with T-mIL12-IRES in both treated and untreated sides on day 13 ($p = 0.043$ and $p = 0.040$ for treated and untreated side, respectively, t-test; Fig. 4b). Between groups, T-mfIL12 was more efficacious than T-mIL12-IRES in the untreated side ($p = 0.018$, two-way ANOVA; Fig. 4b). Data on the tumor growth of individual animals are provided in Supplementary Fig. 3 (Supplementary Data 4).

To confirm that the enhanced efficacy of T-mfIL12 and T-mIL12-IRES over T-01 is due to the immune actions of virus-encoded IL-12, the efficacy was further evaluated in NOG mice, eliminating the effects of adaptive immunity as well as NK activity (Fig. 4c, Supplementary Data 4). When established bilateral subcutaneous Neuro2a tumors reached approximately 5 mm in diameter, T-01, T-mfIL12, T-mIL12-IRES ($1 \times 10^6$ pfu) or mock was inoculated into the left tumors only on days 0 and 3. In the treated side, all three viruses exhibited significant efficacy

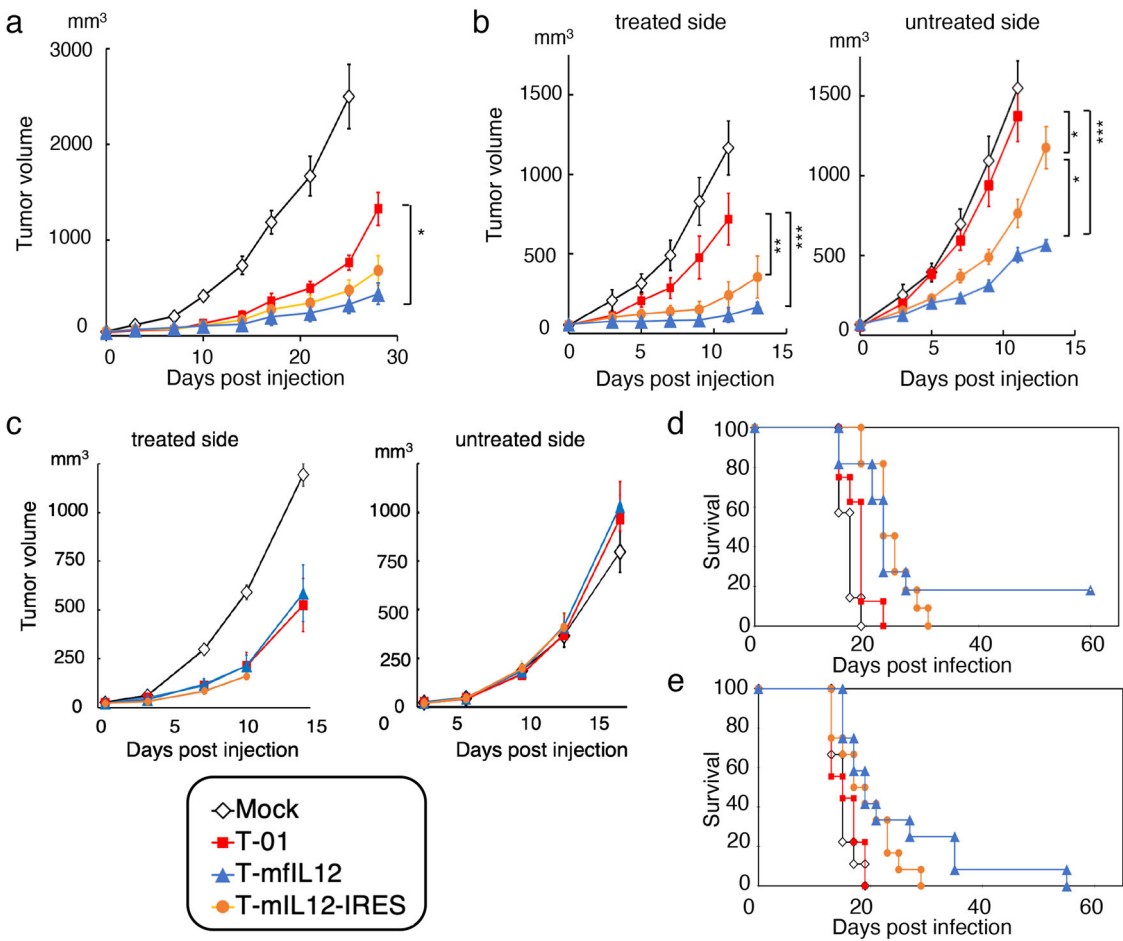

**Fig. 4 In vivo efficacy of T-mfIL12 and T-mIL12-IRES. a, b** and **c** Assessment by subcutaneous tumor growth inhibition. **a** Male C57BL/6 mice harboring unilateral subcutaneous TRAMP-C2 tumors (prostate cancer) were treated with intratumoral inoculations with T-01, T-mfIL12 or T-mIL12-IRES ($5 \times 10^6$ pfu) or mock on days 0 and 3 (n = 6). T-mfIL12 showed the highest efficacy. All three viruses were significantly more efficacious than mock (p < 0.001 for all) (Significance not indicated in the figure). **b** Female A/J mice harboring bilateral subcutaneous Neuro2a tumors (neuroblastoma) were treated with intratumoral inoculations with T-01, T-mfIL12 or T-mIL12-IRES ($5 \times 10^4$ pfu) or mock into the left tumors only on days 0 and 3 (n = 9-12). T-mfIL12 exhibited the highest efficacy in both treated and untreated sides. All three viruses were significantly more efficacious than mock in the treated side ($p = 0.001$ for T-01 and $p < 0.001$ for both T-mfIL12 and T-mIL12-IRES) and T-mfIL12 and T-mIL12-IRES than mock in the untreated side ($p < 0.001$ for both viruses) (Significance not indicated in the figure). **c** NOG mice harboring bilateral subcutaneous Neuro2a tumors were treated with intratumoral inoculations with T-01, T-mfIL12 or T-mIL12-IRES ($1 \times 10^6$ pfu) or mock into the left tumors only on days 0 and 3 (n = 10). T-01, T-mfIL12 and T-mIL12-IRES showed no difference in efficacy in the treated side. The effect of IL-12 expression was completely abolished for both T-mfIL12 and T-mIL12-IRES in both the treated side and the untreated side. Tumor volume = length × width × height × 0.52. Results represent the mean. Bars, standard error of the mean (SEM). *, $p < 0.05$; **, $p < 0.01$; ***, $p < 0.001$; Two-way ANOVA with Bonferroni's multiple comparisons test. **d** and **e** Assessment by survival. Female A/J mice harboring bilateral subcutaneous Neuro2a tumors were treated with intratumoral inoculations with T-01, T-mfIL12 or T-mIL12-IRES, either $5 \times 10^5$ pfu (**d**) or $5 \times 10^4$ pfu (**e**), or mock into the left tumors on days 0 and 3. Animals were sacrificed when the maximum diameter of one of the bilateral tumors reached 22 mm. Both T-mfIL12 and T-mIL12-IRES significantly prolonged the survival of tumor-bearing mice compared with T-01 at both doses (log-rank test). One or two T-mfIL12-treated mice showed a long-term survival for both doses, but there was no significant difference between T-mfIL12 and T-mIL12-IRES.

compared with mock ($p < 0.001$ vs mock for all viruses, two-way ANOVA). However, the effect of IL-12 expression was completely abolished in NOG mice for both T-mfIL12 and T-mIL12-IRES, and there was no difference in efficacy among the three viruses T-01, T-mfIL12 and T-mIL12-IRES. All three viruses had no effect on the contralateral untreated tumors (Fig. 4c).

The efficacy was further evaluated by survival of A/J mice harboring bilateral subcutaneous Neuro2a tumors. When established subcutaneous Neuro2a tumors reached approximately 5 mm in diameter, T-01, T-mfIL12, T-mIL12-IRES or mock was inoculated into the left tumors on days 0 and 3, and mice were sacrificed when one of the tumors reached 22 mm in diameter. Two doses were tested, $5 \times 10^5$ pfu and $5 \times 10^4$ pfu, in two separate experiments. When $5 \times 10^5$ pfu was used, both T-mfIL12 and T-

mIL12-IRES significantly prolonged the survival of tumor-bearing mice compared with T-01 ($p = 0.023$ and $p = 0.002$ for T-mfIL12 and T-mIL12-IRES, respectively, log-rank test; Fig. 4d, Supplementary Data 4). However, only the T-mfIL12 group showed survivors at day 60 (2/11). When $5 \times 10^4$ pfu was used, both T-mfIL12 and T-mIL12-IRES significantly prolonged the survival of tumor-bearing mice compared with T-01 ($p = 0.015$ and $p = 0.046$ for T-mfIL12: and T-mIL12-IRES, respectively, log-rank test; Fig. 4e, Supplementary Data 4). Although there were one or two T-mfIL12-treated mice that showed a long-term survival for both doses, there was no significant difference between T-mfIL12 and T-mIL12-IRES ($p = 0.974$ for $5 \times 10^5$ pfu [Fig. 4d] and $p = 0.265$ for $5 \times 10^4$ pfu [Fig. 4e], log-rank test). Overall results indicate that T-mfIL12 exhibits higher in vivo efficacy than T-mIL12-IRES.

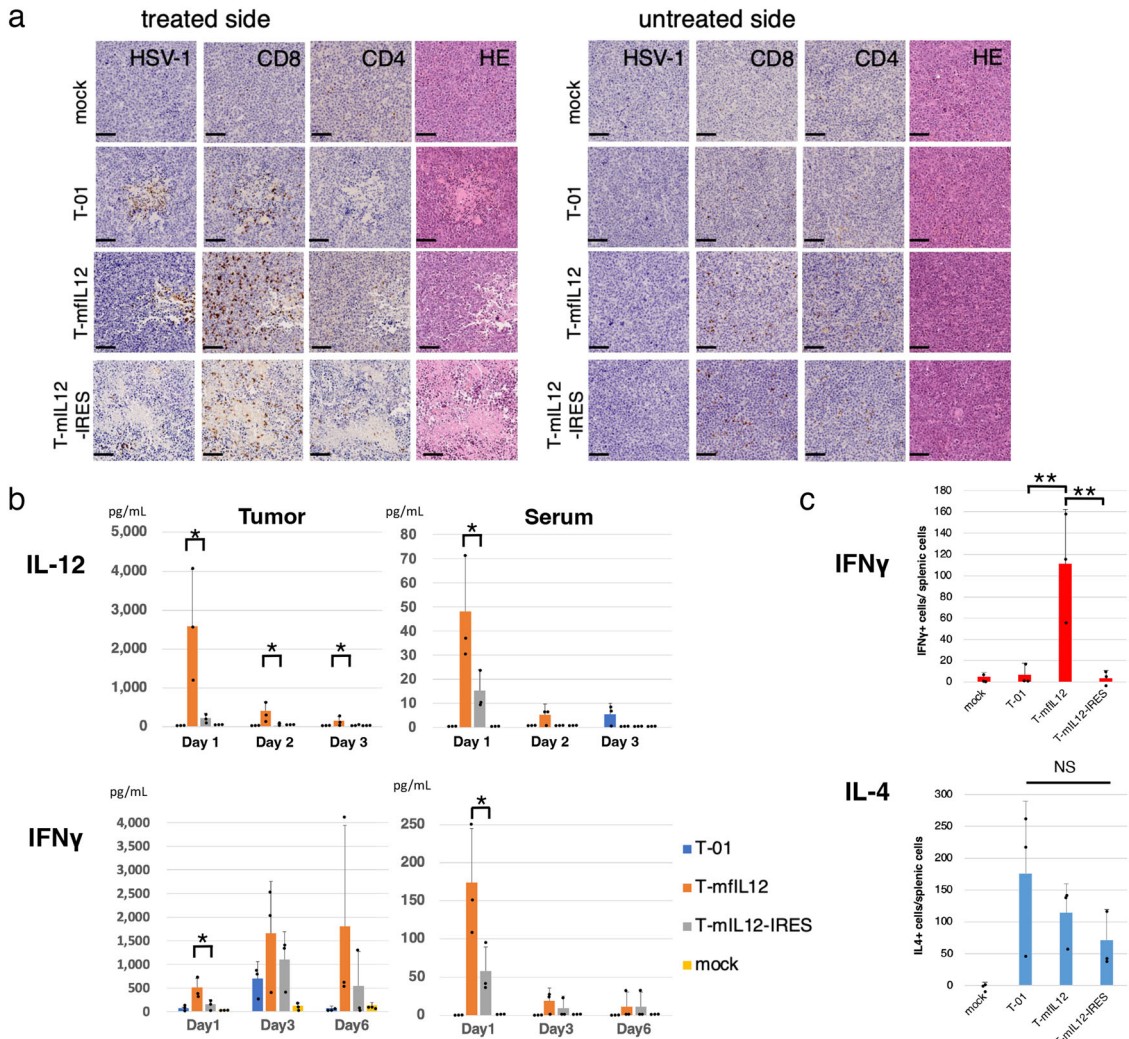

**Fig. 5 Immune responses by T-mfIL12 and T-mIL12-IRES. a** Immunohistochemistry. Bilateral subcutaneous Neuro2a tumors were generated in A/J mice, left tumors only were inoculated with T-mfIL12, T-mIL12-IRES, T-01 ($2 \times 10^5$ pfu) or mock on days 0 and 3, and the tumors were harvested on day 6 (n = 3 per group). An increased infiltration of CD4$^+$ and CD8$^+$ lymphocytes were observed in the tumor for all three viruses, both in the treated and the untreated side, most prominently with T-mfIL12 (Fig. 5a). HSV-1 positive cells were observed in the tumor with all viruses in the treated side, but not in the untreated side. HE, hematoxylin and eosin. Scale bars, 100 μm. **b** In vivo levels of interleukin-12 (IL-12) and Interferon γ (IFNγ). In the unilateral subcutaneous Neuro2a model, established tumors were inoculated with T-01, T-mfIL12, T-mIL12-IRES ($2 \times 10^6$ pfu) or mock, sera and tumor samples were collected on days 1, 3 and 6 (n = 3 per group), and the levels of mouse IL-12 and IFNγ were measured by ELISA. The intratumoral IL-12 levels for T-mfIL12 were significantly higher than those for T-mIL12-IRES at all time points ($p = 0.018$, $p = 0.016$ and $p = 0.046$ for days 1, 3 and 6, respectively). The levels of IL-12 detected from the serum were remarkably lower than those in the tumor. Correlating with the intratumoral IL-12, the serum IL-12 level for T-mfIL12 was higher than that for T-mIL12-IRES on day 1 ($p = 0.047$). The IFNγ levels of T-mfIL12 were significantly higher than those of T-mIL12-IRES both in the tumor and serum on day 1 ($p = 0.014$ and $p = 0.027$, tumor and serum, respectively). **c** Immune responses specific to Neuro2a cells. In the unilateral subcutaneous Neuro2a model, established tumors were inoculated with T-01, T-mfIL12, T-mIL12-IRES ($5 \times 10^4$ pfu) or mock on days 0 and 3, and the spleen was harvested on day 6. By ELISpot assay, splenocytes from T-mfIL12-treated mice showed a significantly higher number of IFNγ release stimulated by Neuro2a cells than those from T-01- and T-mIL12-IRES- treated ones ($p = 0.005$ and $p = 0.004$ vs T-01 and T-mIL12-IRES, respectively). No significant difference in number of IL-4 releasing splenocytes was observed among the three virus-treated groups. The IFNγ or IL-4 releases specific to Neuro2a cells were calculated by subtracting the numbers of Sal/N responding spots from those of Neuro2a responding spots. For **b** and **c**, Graphs show the means. Dots represent individual data. Bars, SD. *, $p < 0.05$; **, $p < 0.01$; NS, not significant; one-way ANOVA with Tukey's multiple comparisons.

We further assessed by immunohistochemistry whether IL-12-expressing viruses promote infiltration of lymphocytes into the tumor. Bilateral subcutaneous Neuro2a tumors were generated in A/J mice, left tumors only were inoculated with T-mfIL12, T-mIL12-IRES, T-01 ($2 \times 10^5$ pfu) or mock on days 0 and 3, and the tumors were harvested on day 6. An increased infiltration of CD4$^+$ and CD8$^+$ lymphocytes were observed in the tumor for all three viruses, both in the treated and the untreated side, more prominently with T-mfIL12 and T-mIL12-IRES than T-01 (Fig. 5a). HSV-1 positive cells were observed in the tumor with

all viruses in the treated side, but none of the viruses caused HSV-1 positivity in the untreated side (Fig. 5a).

**In vivo levels of IL-12 and IFNγ by T-mfIL12 and T-mIL12-IRES.** To assess the in vivo transgene expression and function by T-mfIL12 and T-mIL12-IRES, we measured the in vivo IL-12 and IFNγ levels in the unilateral subcutaneous Neuro2a model. Established tumors were inoculated with T-01, T-mfIL12, T-mIL12-IRES ($2 \times 10^6$ pfu) or mock, and sera and tumor samples

were collected on days 1, 3 and 6 (n = 3 per group). In the tumor, the levels of IL-12 peaked on day 1, measuring 2590 ± 1450 pg/ml and 228 ± 115 pg/ml for T-mfIL12 and T-mIL12-IRES, respectively (mean ± SD), and both IL-12 levels gradually decreased by days 3 and 6 (Fig. 5b, Supplementary Data 5). The intratumoral IL-12 levels on day 0, immediately after virus injections, were 7.7 ± 9.9 pg/ml for T-mfIL12 and not detected for T-mIL12-IRES, therefore the IL-12 detected on day 1 and after resulted from the IL-12 expression by the viruses. The levels of intratumoral IL-12 for T-mfIL12 were significantly higher than those for T-mIL12-IRES at all time points ($p = 0.018$, $p = 0.016$ and $p = 0.046$ for days 1, 3 and 6, respectively, one-way ANOVA). The levels of IL-12 detected from the serum were remarkably lower (approximately 2 log lower) than those in the tumor. Correlating with the intratumoral IL-12, the serum IL-12 level for T-mfIL12 was higher than that for T-mIL12-IRES on day 1 ($p = 0.047$, one-way ANOVA). IFNγ was detected in the tumor from day 1 for all three viruses, which increased by day 3 (Fig. 5b). IFNγ was detected from the serum with IL-12-expressing viruses only, which peaked on day 1. The IFNγ levels for T-mfIL12 were significantly higher than those for T-mIL12-IRES both in the tumor and serum on day 1 ($p = 0.014$ and $p = 0.027$, tumor and serum, respectively, one-way ANOVA).

**Specific antitumor immune responses by T-mfIL12 and T-mIL12-IRES.** To assess the specific antitumor immune responses elicited by T-mfIL12 and T-mIL12-IRES, we performed ELISpot assays in the unilateral subcutaneous Neuro2a model. Established tumors were inoculated with T-01, T-mfIL12, T-mIL12-IRES ($5 \times 10^4$ pfu) or mock on days 0 and 3, and the spleen was harvested on day 6. By ELISpot assay, splenocytes from T-mfIL12-treated mice showed a significantly higher number of IFNγ release stimulated by Neuro2a cells than those from T-01- and T-mIL12-IRES- treated ones ($p = 0.005$ and $p = 0.004$ vs T-01 and T-mIL12-IRES, respectively, one-way ANOVA; Fig. 5c, Supplementary Data 5). Such responses were not observed with SaI/N cells, control cells derived from A/J mouse strain (Supplementary Fig. 4, Supplementary Data 5). No significant difference in number of IL-4 releasing splenocytes was observed among the three virus-treated groups (Fig. 5c, Supplementary Fig. 5, Supplementary Data 5).

**Comparison of in vivo efficacy of T-mfIL12 and intratumoral recombinant IL-12 administration.** To evaluate the benefit of expressing IL-12 as an armed oncolytic HSV-1, we compared the efficacy of T-mfIL12 with that of direct intratumoral injections with recombinant murine IL-12 (rIL-12) in A/J mice harboring bilateral subcutaneous Neuro2a tumors (Fig. 6, Supplementary Data 6). It has been reported using immunocompetent mice bearing poorly immunogenic subcutaneous tumors that 500 ng of rIL-12 can cause tumor growth inhibition when injected intratumorally[34]. From the above experiment of Fig. 5b, the highest intratumoral IL-12 level by intratumoral injection with $2 \times 10^6$ pfu of T-mfIL12 was calculated to be 45.7 ng/tumor ($30.1 \pm 9.6$, mean ± SEM). Further, when subcutaneous Neuro2a tumors were treated with intratumoral injections with $5 \times 10^4$ pfu of T-mfIL12, the highest intratumoral IL-12 level was measured to be 0.668 ng/tumor ($0.238 \pm 0.147$, n = 4). Hence, for rIL-12, we tested three different doses, 500 ng, 50 ng and 1 ng. When established tumors reached approximately 5 mm in diameter, rIL-12, T-01 ($5 \times 10^4$ pfu) without or with rIL-12, T-mfIL12 ($5 \times 10^4$ pfu) or mock was inoculated into the left tumors only on days 0 and 4 (n = 10 per group). When the dose of 500 ng was used for rIL-12, rIL-12, T-01, T-01+rIL-12 and T-mfIL12 were all significantly more efficacious than mock in the treated side ($p < 0.001$ vs mock for all, two-way ANOVA; Fig. 6a). In the

untreated side, rIL-12 alone showed no significant antitumor effect compared with mock, whereas T-01+rIL-12 and T-mfIL12 showed a significantly higher efficacy than mock ($p = 0.039$ and $p < 0.001$ vs mock, respectively, two-way ANOVA). Further, in the untreated side, T-mfIL12, but not T-01+rIL-12, was significantly more efficacious than rIL-12 alone ($p = 0.003$, two-way ANOVA). Similar results were obtained when the dose of 50 ng was used for rIL-12 (Fig. 6b). When the dose of 1 ng was used for rIL-12, the dose representing the intratumoral IL-12 level treated with T-mfIL12 used in these experiments, rIL-12 alone showed no significant efficacy in both treated and untreated sides, and T-mfIL12, but not T-01 + rIL-12, was significantly more efficacious than rIL-12 alone in the treated side ($p < 0.001$, two-way ANOVA; Fig. 6c). These results implicate that virus-encoded IL-12 acts more effectively than recombinant IL-12 on both local and systemic antitumor immunity.

**Discussion**

In the present study, we compare the efficacy of two types of armed oncolytic HSV-1 that express the same molecule, murine IL-12, regulated by the same immediate-early CMV promoter, but by different methods: T-mfIL12 expresses murine IL-12 as a single, active fusion peptide with the p35 and p40 subunits linked by bovine elastin motifs, and T-mIL12-IRES co-expresses the p35 subunit and p40 subunit, with an IRES sequence inserted between the two subunit genes. We show that a fusion-type expression is significantly more efficient in producing IL-12 in various cell lines than co-expression of two subunits, whereas the expression methods do not affect the replication capabilities and cytopathic effects of armed oncolytic HSV-1. Although both T-mfIL12 and T-mIL12-IRES show a significant augmentation of efficacy compared with the control virus T-01, T-mfIL12 exhibits a significantly higher efficacy than T-mIL12-IRES in vivo in two different syngeneic mouse tumor models, reflecting a higher amount of active IL-12 expressed by T-mfIL12 than T-mIL12-IRES. The function of virus-encoded IL-12 is confirmed to be immune-mediated, as the enhanced efficacy is abolished in NOG mice. A significantly higher amounts of IL-12 are detected in T-mfIL12-treated tumors than T-mIL12-IRES-treated ones, leading to stronger antitumor immune responses as evidenced by IFNγ measurement, ELISpot assay and immunohistochemistry in the poorly immunogenic, Neuro2a tumor model in HSV-1 sensitive A/J mice. This tumor model was used in this study, because we consider it one of the most suitable models to evaluate the functions of immunomodulatory payloads of oncolytic HSV-1.

In addition to direct cell killing, oncolytic HSV-1 elicits a tumor-specific immune response[35–37], and one approach to enhance the efficacy of oncolytic HSV-1 is to arm the virus with an immunostimulatory molecule. Several studies have evaluated different immunostimulatory transgenes by arming each of them in the same oncolytic HSV-1 backbone. The oncolytic HSV-1 expressing murine IL-12 (NV1042) showed higher efficacy than the one expressing murine GM-CSF (NV1034) in mice with subcutaneous squamous cell carcinoma[19]. NV1042-treated mice had a higher rate of rejecting re-challenged tumor cells than NV1034-treated ones[19]. In two mouse prostate cancer models, NV1042, but not NV1034, demonstrated higher efficacy than the control HSV-1[38]. We have created three oncolytic HSV-1 armed with murine soluble B7-1, murine IL-12, or murine IL-18 (designated vHsv-B7-1-Ig, vHsv-IL-12, and vHsv-IL-18, respectively) using the same HSV-1 backbone with deletions in the γ34.5 and ICP6 genes[20]. Whereas the triple combination of vHsv-B7.1-Ig, vHsv-IL-12, and vHsv-IL-18 exhibited greater efficacy than any single virus or any combination of two viruses in A/J mice harboring subcutaneous Neuro2a tumors, vHsv-IL-12

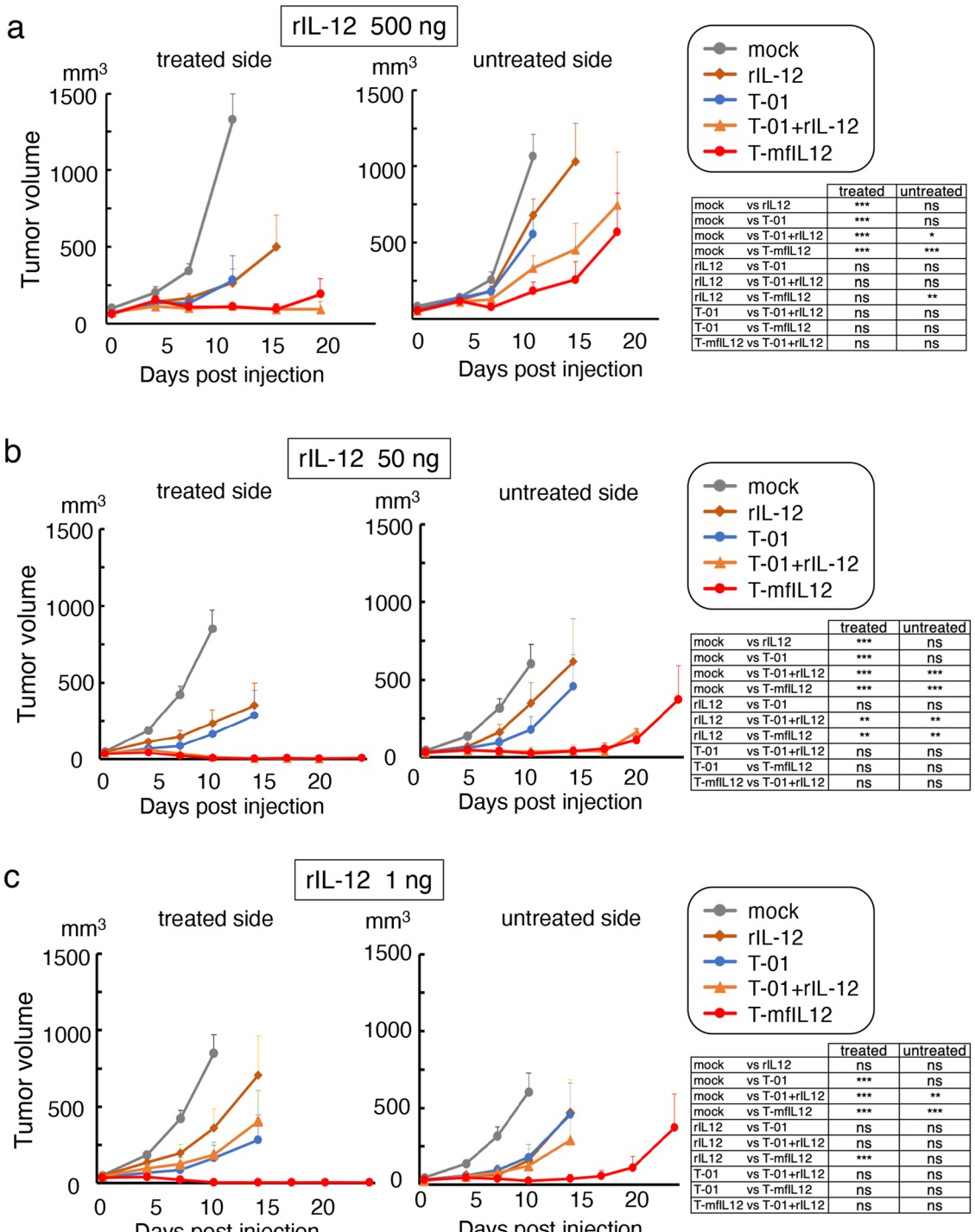

**Fig. 6 Comparison of in vivo efficacy of T-mfIL12 and direct intratumoral injection with recombinant interleukin-12 (rIL-12).** In the bilateral subcutaneous Neuro2a model, rIL-12, T-01 ($5 \times 10^4$ pfu) without or with rIL-12, T-mfIL12 ($5 \times 10^4$ pfu) or mock was inoculated into the left tumors only on days 0 and 4 (n = 10 per group). Three different doses were used for rIL-12; 500 ng (**a**), 50 ng (**b**) and 1 ng (**c**). The dose 50 ng represents the intratumoal IL-12 level treated with T-mfIL12 at $2 \times 10^6$ pfu and 1 ng represents that at $5 \times 10^4$ pfu, the T-mfIL12 dose used in these experiments. **a** When the dose of 500 ng was used for rIL-12, rIL-12, T-01, T-01+rIL-12 and T-mfIL12 were all significantly more efficacious than mock in the treated side ($p < 0.001$ vs mock for all). In the untreated side, rIL-12 alone showed no significant antitumor effect compared with mock, whereas T-01+rIL-12 and T-mfIL12 showed a significantly higher efficacy than mock ($p = 0.039$ and $p < 0.001$ vs mock, respectively). Further, in the untreated side, T-mfIL12, but not T-01+rIL-12, was significantly more efficacious than rIL-12 alone ($p = 0.003$). **b** Results similar to **a** were obtained when the dose of 50 ng was used for rIL-12. **c** When the dose of 1 ng was used for rIL-12, rIL-12 alone showed no significant efficacy in both treated and untreated sides, and T-mfIL12, but not T-01+rIL-12, was significantly more efficacious than rIL-12 alone in the treated side. Tumor volume = length × width × height × 0.52. Results represent the mean. Bars, standard error of the mean (SEM). *, $p < 0.05$; **, $p < 0.01$; ***, $p < 0.001$; ns, not significant; Two-way ANOVA with Bonferroni's multiple comparisons test.

showed the highest efficacy among the three viruses when used alone. These studies indicate that IL-12 is a good choice of transgene so far, if one is to arm an oncolytic HSV-1 with a single immunostimulatory transgene. Systemic administration of IL-12 has shown to cause severe toxicity, including death, in clinical trials[39,40]. Therefore, a local expression of IL-12 using oncolytic HSV-1 as a delivery tool is a reasonable approach also from a safety aspect[41]. Several clinical trials of IL-12 gene therapy have been performed lately using adenovirus vectors[42,43].

Besides those mentioned above, IL-12 has been tested as a transgene for arming other oncolytic HSV-1 for preclinical as well as clinical purposes. G47Δ-mIL12 has the murine IL-12 gene in the G47Δ backbone, similarly to T-mfIL12, but was created using our previous G47Δ-BAC system[21,31]. In immunocompetent mouse glioblastoma stem cell model, G47Δ-mIL12 attacked murine glioblastoma cells, increased release of IFNγ, inhibited angiogenesis, and reduced the number of regulatory T cells in the tumor[21]. The antitumor effect of G47Δ-mIL12 in combination with anti-PD-1 and anti-CTLA-4 antibodies was dependent on CD4[+] and CD8[+] T cells and macrophages[44,45]. While temozolomide chemotherapy antagonized G47Δ-mIL12[46], G47Δ-mIL12 combined with systemic VEGFR tyrosine kinase inhibitor[47] or angiostatin-expressing G47Δ[48] showed enhanced efficacy in mouse glioblastoma model. Another first-generation, γ34.5-deficient oncolytic HSV-1 armed with murine IL-12, M002, showed higher efficacy than the parental virus and G207 in preclinical brain tumor models[22,49]. M032 was then constructed using the same backbone and strategy as M002, but expressing human IL-12 instead of murine, for the purpose of performing a clinical trial in patients with high grade glioma[50]. Intracerebral administration with M032 (up to $10^8$ pfu) was found safe in studies with non-human primates[51]. M032 is used in a phase 1 study in patients with recurrent malignant glioma (NCT02062827). Other approaches to arm oncolytic HSV-1 with IL-12 have been reported, including insertion of the human IL-12 gene in the ICP47- and ICP34.5-deleted HSV-1 backbone (Δ47/Δ34.5/IL12)[52] and arming fully-virulent, HER2-retargeted oncolytic HSV-1 with murine IL-12 (R-115)[53,54].

NV1042 or G47Δ-mIL12 used the murine IL-12 gene that was expressed as a single fusion protein[19,21], whereas M002, M032, Δ47/Δ34.5/IL12 and R-115 used the murine or human IL-12 gene in which the p35 and p40 subunit genes were separated by IRES and driven by the early growth response-1 promoter or the CMV promoter thus co-expressed as two proteins. IL-12 is species specific, therefore, after obtaining preclinical data using oncolytic HSV-1 armed with murine IL-12, a similar but newly constructed oncolytic HSV-1 armed with human IL-12 is required to start a clinical trial. Because clinical development requires time, effort and money, it is difficult to turn back or change course once it starts moving forward. Therefore, it is important to consider whether the oncolytic virus to be clinically developed is optimal in all aspects, including safety, efficacy, manufacturing, stability, clinical practicality, regulatory hurdles and environmental impact. As one of such aspects, this study evaluated the optimal method of expressing IL-12 as a payload of oncolytic HSV-1. The results indicate that expression of the IL-12 gene as a single fusion peptide is remarkably superior to co-expressing the two subunits. As a result of this study, we used the T-BAC system, an improved version of G47Δ-BAC system[31], that allows a rapid and precise insertion of a desired transgene into the deleted ICP6 locus of G47Δ, to generate a G47Δ-based oncolytic HSV-1 that expresses human IL-12 as a functional fusion peptide (T-hIL12). T-hIL12 has been well characterized, including the functions of expressed human IL-12, and T-hIL12 is currently used in ongoing, investigator-initiated, phase 1/2 clinical trial in patients with malignant melanoma in Japan (jRCT2033190086).

## Data availability
The data that support the findings of this study are available from the corresponding author on reasonable request. All source data are provided as Supplementary Data.

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

## Acknowledgements

This work was supported in part by research grants from the Ministry of Education, Culture, Sports, Science and Technology (MEXT) of Japan (grant no. 09028017 for T.T.), the Japan Agency of Medical Research and Development (AMED) (grant no. JP15cm0106015 for T.T.), and Grant-in-Aid for Scientific Research from Japan Society for the Promotion of Science (JSPS Kakenhi) (grant no. 16712019 for H.F.). We thank Dr. Yasushi Ino for his helpful advice, Sayaka Kanaami, Eri Ozaki, Kyoko Saiga, Rumi Takashima for their technical assistance, and Dr. Jeffery Green for providing us with Pr14-2 cells.

## Author contributions

Conceived and designed the experiments, T.T.; performed the experiments, H.F., Y.T.S., J.H., M.I.; analyzed the data, H.F., M.I., T.T.; contributed reagents/materials/analysis tools, H.F., T.T.; supervision, T.T.; wrote the paper, H.F., T.T.

## Competing interests
The authors declare no competing interests.

## Additional information

**Peer review information** : *Communications Medicine* thanks E. Antonio Chiocca and the other, anonymous, reviewer(s) for their contribution to the peer review of this work. Peer reviewer reports are available.

