## [Peer Review File · Communications Medicine]

Reviewers' comments:

Reviewer #1 (Remarks to the Author):

Todo and co-workers report on oncolytic herpes virus derivatives encoding interleukin-12 (IL-12).

Specifically, they compare a variant encoding an IL-12 fusion protein of the IL-12 subunits p30 and p40 with a variant including an IRES sequence.

The herpes virus G47delta they used as backbone has been approved for treatment of malignant glioma in Japan, highlighting potential translational relevance.

This work builds on a body of previous publications on oncolytic herpes viruses.

Several previous studies have investigated herpes viruses encoding IL-12, demonstrating improved efficacy compared to parental viruses without an additional immunomodulatory transgene, e.g.,

Parker et al. 2000, DOI: 10.1073/pnas.040557897

Wong et al. 2001, DOI: 10.1089/10430340150218396

Wong et al. 2004, DOI: 10.1158/1078-0432.CCR-04-0081

Cheema et al. 2013, DOI: 10.1073/pnas.1307935110

The finding that expressing IL-12 as a fusion protein is beneficial is not surprising, since previous publications have established this (e.g., Lieschke et al. 1997, DOI: 10.1038/nbt0197-35).

Here, the authors describe the production of the two novel virus variants. Viral replication is assessed by progeny titration 48 post infection.

Compared to parental G47delta, there is a moderate reduction, but there is no significant difference compared to a control virus harboring an empty transgene cassette.

ELISA measurements of cell culture supernatants showed that the variant encoding the fusion protein yields significantly higher levels of IL-12 compared to the IRES variant.

Direct cytotoxic effects were assessed by counting viable cells after infection with IL-12 encoding variants or control virus, showing no differences.

In vivo studies were performed in two immunocompetent mouse models.

In a subcutaneous TRAMP-C2 model, treatment with virus delayed tumor progression, with a slight benefit for IL-12 encoding variants.

In a bilateral Neuro2a model, the IL-12 encoding variants appeared to also slow tumor growth of both treated in non-treated tumors, indicating possible abscopal effects.

Both higher and lower doses of the IL-12 variants prolonged survival in the subcutaneous Neuro2a model. There were no statistically significant differences between the two variants. Long-term survival (1 - 2 per experiment) was only achieved by treatment with the fusion protein variant.

Overall novel immunomodulatory oncolytic viruses are an interesting and clinically relevant area of investigation.

However, the novelty of this approach is limited and the data as it stands appears preliminary. Moreover, the advantages of the IL-12 fusion protein variant are small

and since all experiments were performed in murine model systems it is unclear what can be expected in a clinical treatment setting.

My specific comments are as follows:

- It seems the final constructs were only verified by endonuclease digest (lines 166-167). Given that mutations may occur during virus generation (lines 132-133), it would be important to verify the constructs, at least the transgene cassettes, by sequencing. This would also help to exclude that mutations are the cause for any differences between the variants.
- In this study an advanced BAC system was used for virus construction. For non-specialists it would be helpful to briefly state the specific advantages of this system as compared to previous approaches.
- Statistics: In many instances, p values are noted without indicating the respective test (e.g., line 235, line 241, line 266, line 268, line 514, line 517). In several sentences, the term "average" is used (e.g., lines 175 and 186). Preferably, the statistical term such as "arithmetic mean" should be used.
- For viral replication, viral progeny is determined at a single timepoint. Growth kinetics/growth curves would be helpful to detect more subtle difference in viral replication, which may however be relevant in vivo in modestly permissive tumors.
- IL-12 expression is quantified by ELISA. However, data on transgene function, e.g. lymphocyte activation by virus-encoded IL-12, is lacking.
- The authors claim that T-mfIL-12 was significantly more efficacious than T-mIL12-IRES. However, this is based on selected datapoints, such as tumor volume on day 25 in the TRAMPC2 model. Overall there does not appear to be a meaningful difference.
- In the Neuro2a model, the control virus has barely any effect. What is the benefit of virus-encoded IL-12 compared to direct intratumoral injection of IL-12? Potential advantages should be backed up with data.
- No data on transgene expression and target engagement in vivo is presented. What intratumoral levels of IL-12 are reached after treatment with the respective virus variants? What effects does treatment have on anti-tumor immunity? The bilateral flank model implies there may be an abscopal effect. Can tumor-specific immune responses be detected, e.g. by ELISpot?
Do the IL-12 encoding viruses promote tumor infiltration with immune cells or activation of tumor-infiltrating lymphocytes? Data on such mechanisms of action is needed to assess the therapeutic potential of the novel viruses.

- What are systemic levels of IL-12 after treatment with the virus variants? This is an important safety aspect with respect to clinical application.
- In the bilateral model, was spread of the virus to the non-treated tumor excluded?
- All data presented comes from mouse cell lines and mouse models. At the end of the discussion, it is stated that a phase 1/2 clinical trial with T-hIL-12 is ongoing. It would be very interesting to see data on the human IL-12 variant virus such as replication and cytotoxicity in human tumor cells, IL-12 expression levels, IL-12 function.
- For the in vivo data, it would be helpful to show individual tumor growth curves for all animals, also beyond day 30/ day 15.

Minor points:

- Line 183: Why was temperature shifted to 39.5°C for the ELISA sampling? Please explain.
- Lines 220 - 221: Instead of "front" and "back", terms such as "upstream" and "downstream" should be used to describe genome sequences.
- Figure 1: Perhaps indicate hours post infection instead of days

Reviewer #2 (Remarks to the Author):

Fukuhara et al. report on the modification of G47 Δ which is a triple-mutated oncolytic herpes simplex virus type 1 (HSV-1) via the expression of murine IL-12.

The authors explored two modalities to accomplish this (i.e., via the creation of T-mfIL12 which expresses murine IL-12 as a fusion peptide, with the genes of two subunits [p35 and p40] linked by bovine elastin motifs, and via the creation of T-mIL12-IRES which co-expresses the subunits, with the two genes separated by an internal ribosome entry site [IRES] sequence).

In so doing, Fukuhara et al examined the expression of functional IL-12 via ELISA, replication of parent viruses and ultimately in vitro/in vivo cytotoxicity/efficacy (using a neuroblastoma and prostate models).

While this is a solid contribution to the field, the role of IL-12 in oHSVs has been well defined and the findings while very important may be more incremental in nature.

A few pertinent suggestions for the authors are highlighted below as they consider revising/resubmission of the manuscript.

Major points:

1. The authors continue to report that G47delta was successful in a Phase 2 trial; aside from an abstract published in 2019 (PMCID: PMC6847570) this reviewer is unaware of a formal

publication reporting on the trial. I would suggest the authors discuss this and/or provide the Japanese clinical trial number in future versions of the manuscript.

2. If the overall guise of this paper is to discuss/explore the preclinical rationale for a human IL-12 fusion protein prior to beginning clinical trials in malignant melanoma as referenced, why employ the preclinical models chosen (i.e., neuroblastoma and prostate) as opposed to melanoma? Further the folding of the human version of the fusion protein should be explored given the distinct differences between human and murine versions of IL-12 protein.

3. The decreased viral yields noted in the IL-12 variants vs the parent G47delta virus should be explored in detail; next generation sequencing may be needed to confirm genomic stability given the insertions (w/ the understanding that GC rich regions within oHSVs may be difficult to cover).

4. Efficacy of survival in a xenograft model should also be explored removing the potential confounder of IL-12/adaptive immunity. This would ensure that replication is equivalent in vivo given the novel transgene insertions.

Minor points:

1. There are a litany of grammatical errors throughout the manuscript that should be addressed; style and flow should also be improved throughout the paper (this may be accomplished via engagement of a native English speaker).

2. Figure 3 is referenced in Results prior to Figure 2; the authors should therefore realign the flow of the figures.

3. The authors should explore expression of p35 vs p40 in the IRES construct to ensure 1:1 levels of transcription/translation; P2A and T2A domains might also be considered as opposed to IRES moving forward.

4. The M032 clinical trial (<https://clinicaltrials.gov/ct2/show/NCT02062827>) should be referenced within the discussion.

5. There have also been published clinical trials of IL12 gene therapy for cancer in the last 3-4 years using adenoviral vectors. These should also be cited since it is relevant to the paper

6. Can the authors comment on the immunogenicity of the bovine linker? Do they anticipate reactions in patients given repeat dosing/exposures?

Reviewer #3 (Remarks to the Author):

In this paper they have developed a triple-mutated oncolytic HSV-1 armed with human IL-12. The oncolytic virus they have chosen to work with is currently used in phase 1/2 trial for

malignant melanoma. They have created two viruses using two approaches to express IL-12: T-mfIL12 expresses murine IL-12 as a fusion peptide, with the genes of two subunits (p35 and p40) linked by bovine elastin motifs, and T-mIL12-IRES co-expresses the subunits, with the two genes separated by an internal ribosome entry site (IRES) sequence. Their claim is the fusion-type expression of IL-12 is superior to co-expression of separate subunits, and results in higher production of functional IL-12 molecules.

The use of OVAs expressing IL-12 is exciting as this overcomes the systemic toxicity of IL-12 given as a cytokine therapy. The generation of IL-12 armed HSV as well as other viruses has been attempted and published in the past. For example, Haghighi et al, inserted IL-12 in the ICP34.5 deleted HSV-1 backbone to improve the oncolytic properties and provide an immune-stimulatory effects in colorectal cancer cells.

<https://doi.org/10.1016/j.micpath.2021.105164> . They used pCDNA3.1+ CMV-hIL12 containing the p40 and p35 subunits of human IL-12. More recent studies using oncolytic HSV armed with IL-12 were shown to have vaccine like properties [doi:10.1371/journal.ppat.1007209](https://doi.org/10.1371/journal.ppat.1007209).

There are many other studies, as well as other genes that have been inserted into HSV for experimental cancer research.

There is also some confusion regarding the use of this virus in clinical trials, G47 Δ was approved as a new drug for malignant glioma in Japan but the choice of wording in the abstract and lay summary suggests the IL-12 armed virus is currently in clinical trials for melanoma. Where is this trial? Are they recruiting?

The authors have performed some elegant experiments introducing IL-12 using a few different approaches for generating their new viruses, they have published on this previously and used a similar system with modifications. They have also shown some convincing data with the viruses they have developed. However, given all the published studies of HSV-IL-12, so I don't believe this article has novelty nor that it will influence thinking in the field. Some other concerns about the paper.

More clarity regarding the choice of the intratumoural dose and why 2 doses of HSV were given is needed. Also why the tumours were grown to be so large. The paper also felt short in the analysis of the mice. It was disappointing that no immune analysis was carried out to determine how the IL-12 viruses changed the tumour microenvironment especially as this was in fully immunocompetent mice. Was there an increase in IL-12 secretion in cell cultures following infection or in the mouse tumours following injection. This may have yielded more exciting data and determined if the methods of virus production produced effective viruses.

Statistical analysis is appropriate but should have been described in the figure legends.

The work is described clearly and a researcher should be able to reproduce the work.

Responses to Reviewers

Detailed responses to the criticisms and requests are listed below. Revised figures, sentences and words are shown in red.

Responses to Reviewer #1

General comments:

Todo and co-workers report on oncolytic herpes virus derivatives encoding interleukin-12 (IL-12). Specifically, they compare a variant encoding an IL-12 fusion protein of the IL-12 subunits p30 and p40 with a variant including an IRES sequence.

The herpes virus G47delta they used as backbone has been approved for treatment of malignant glioma in Japan, highlighting potential translational relevance. This work builds on a body of previous publications on oncolytic herpes viruses. Several previous studies have investigated herpes viruses encoding IL-12, demonstrating improved efficacy compared to parental viruses without an additional immunomodulatory transgene, e.g.,

Parker et al. 2000, DOI: 10.1073/pnas.040557897

Wong et al. 2001, DOI: 10.1089/10430340150218396

Wong et al. 2004, DOI: 10.1158/1078-0432.CCR-04-0081

Cheema et al. 2013, DOI: 10.1073/pnas.1307935110

The finding that expressing IL-12 as a fusion protein is beneficial is not surprising, since previous publications have established this (e.g., Lieschke et al. 1997, DOI: 10.1038/nbt0197-35).

Here, the authors describe the production of the two novel virus variants. Viral replication is assessed by progeny titration 48 post infection. Compared to parental G47delta, there is a moderate reduction, but there is no significant difference compared to a control virus harboring an empty transgene cassette. ELISA measurements of cell culture supernatants showed that the variant encoding the fusion protein yields significantly higher levels of IL-12 compared to the IRES variant. Direct cytotoxic effects were assessed by counting viable cells after infection with IL-12 encoding variants or control virus, showing no differences. In vivo studies were performed in two immunocompetent mouse models. In a subcutaneous TRAMP-C2 model, treatment with virus delayed tumor progression, with a slight benefit for IL-12 encoding variants. In a bilateral Neuro2a model, the IL-12 encoding variants appeared to also slow tumor growth of both treated in non-treated tumors, indicating possible abscopal effects. Both higher and lower doses of the IL-12 variants prolonged survival in the subcutaneous Neuro2a model. There were no statistically significant differences between the two variants. Long-term survival (1 - 2 per experiment) was

only achieved by treatment with the fusion protein variant.

Overall novel immunomodulatory oncolytic viruses are an interesting and clinically relevant area of investigation. However, the novelty of this approach is limited and the data as it stands appears preliminary. Moreover, the advantages of the IL-12 fusion protein variant are small and since all experiments were performed in murine model systems it is unclear what can be expected in a clinical treatment setting.

Responses:

We thank the reviewer for the general comments. This paper focuses on evaluating the optimal methods for expressing IL-12 as a payload for G47 Δ -based oncolytic HSV-1 by comparing different types of IL-12 transgene. As far as we know, there has never been a study that compared the expression of IL-12 as a fusion peptide with the co-expression of the two subunit genes utilizing IRES using the same oncolytic HSV-1 backbone and for the purpose of selecting the optimal method to encode IL-12 for clinical development. Thanks to the reviewer's critical comments and requests, a significant amount of data both in vitro and in vivo are now added to this paper. The results show that T-mfIL12 (a fusion-type variant) not only exhibits higher amounts of active IL-12 in vitro and higher efficacy in the two immunocompetent mouse tumor models but also shows a significantly higher intratumoral expression of functional IL-12, causing stronger stimulation of specific antitumor immune responses, than T-mIL12-IRES (co-expression type variant). IL-12 is species specific, therefore, we agree that, as with any other transgenes coding species-specific peptides, the preclinical data acquired in animal models can only be attested in clinical trials, one of which is ongoing in patients with malignant melanoma using T-hIL12, the human version of T-mfIL12. Point by point responses to the reviewer's specific comments are listed below.

Major point 1:

It seems the final constructs were only verified by endonuclease digest (lines 166-167). Given that mutations may occur during virus generation (lines 132-133), it would be important to verify the constructs, at least the transgene cassettes, by sequencing. This would also help to exclude that mutations are the cause for any differences between the variants.

Response:

In compliance with the reviewer's comments, we performed sequencing of the transgene cassettes in addition to endonuclease digestion to verify the structure of the viral DNA. We confirmed that the sequences of the p35 subunit and the p40 subunit of T-mfIL12 were identical to those of T-mIL12-IRES. **Supplementary Figure 2** is newly added.

The following statement is added to the Methods, “Virus construction” section (p.11, 1st paragraph):

Transgene cassettes of the recombinant viruses were sequenced (primers: 5'-CGCAAATGGGCGGTAGGCGTG-3', 5'-TAGAAGGCACAGTCGAGG-3', 5'-ACCCGCCAAGAAGACTTGCAG-3', 5'-GGATCGGACCCTGCAGGGAAC-3'). Primers were designed to sequence the nucleotides between the CMV promoter and polyA.

The following statement is added to the Results, “Construction of T-mfIL12 and T-mIL12-IRES” (p.17, 1st paragraph, last sentence).

Transgene cassettes of T-mfIL12 and T-mIL12-IRES were sequenced to confirm that the sequences of the p35 subunit and the p40 subunit of T-mfIL12 were identical to those of T-mIL12-IRES (Supplementary Figure 2).

Major point 2:

In this study an advanced BAC system was used for virus construction. For non-specialists it would be helpful to briefly state the specific advantages of this system as compared to previous approaches.

Response:

Conventionally, recombinant HSV-1 have been constructed using homologous recombination techniques, which required time and effort consuming processes of selecting and confirming the structures of a desired recombinant that occurred at a very low probability, because of the large genome size of HSV-1. In order to eliminate the element of chance as much as possible, we previously constructed the G47Δ-BAC system using bacterial artificial chromosome (BAC) and two recombinases, allowing the development of multiple armed oncolytic HSV-1 viruses in parallel, swiftly and precisely, using G47Δ as the backbone³¹. However, G47Δ-BAC products were later found to have lower replication capabilities than G47Δ, presumably due to unknown mutations in the G47Δ-BAC plasmid. We therefore reconstructed the system entire system under the same concept, but with added modifications including useful multiple cloning sites. This T-BAC system allows production of G47Δ-derived armed oncolytic HSV-1 with replication capabilities comparable to G47Δ and with desired transgenes.

The above is now described in the Methods, “Virus construction” section (p.8, 2nd paragraph):

Conventionally, recombinant HSV-1 have been constructed using homologous recombination techniques, which required time and effort consuming processes of selecting and confirming the

structures of a desired recombinant that occurred at a very low probability, because of the large genome size of HSV-1. In order to eliminate the element of chance as much as possible, we previously constructed the G47Δ-BAC system using bacterial artificial chromosome (BAC) and two recombinases, allowing the development of multiple armed oncolytic HSV-1 viruses in parallel, swiftly and precisely, using G47Δ as the backbone³¹. However, G47Δ-BAC products were later found to have lower replication capabilities than G47Δ, presumably due to unknown mutations in the G47Δ-BAC plasmid. We therefore reconstructed the entire system under the same concept, but with added modifications including useful multiple cloning sites. This T-BAC system allows production of G47Δ-derived armed oncolytic HSV-1 with replication capabilities comparable to G47Δ and with desired transgenes.

Major point 3:

Statistics: In many instances, p values are noted without indicating the respective test (e.g., line 235, line 241, line 266, line 268, line 514, line 517). In several sentences, the term "average" is used (e.g., lines 175 and 186). Preferably, the statistical term such as "arithmetic mean" should be used.

Response:

In compliance with the reviewer's comments, we indicated the method of statistical analysis used, either t-test (Student's t-test), one way ANOVA (one-way ANOVA with Tukey's multiple comparisons test) or two-way ANOVA (two-way ANOVA with Bonferroni's multiple comparisons test). We changed the term "average" to "arithmetic mean" in pages 13, 17 and 18.

Major point 4:

For viral replication, viral progeny is determined at a single timepoint. Growth kinetics/growth curves would be helpful to detect more subtle difference in viral replication, which may however be relevant in vivo in modestly permissive tumors.

Response:

In accordance with the reviewer's advice, we performed an additional experiment to determine the number of viral progenies at multiple timepoints in Vero cells. The time course for the viral yields of T-mfIL12 was comparable to that of T-mIL12-IRES. The result is shown in newly added **Figure 2C**.

The following description is added in the Methods, “Replication assay and cytopathic effect studies” section (p.12, 1st paragraph):

In a separate experiment, Vero cells were infected with G47Δ, T-01, T-mfIL12 or T-mIL12-IRES in duplicate wells at an MOI of 0.01, the progeny virus was recovered 0 h, 6 h, 24 h and 48 h after infection, and titrated by plaque assay.

The following description is further added in the Results, “In vitro characteristics of T-mfIL12 and T-mIL12-IRES” section (p.18, 1st paragraph):

The time course (0h, 6h, 24h and 48h) for the viral yields of T-mfIL12 in Vero cells was comparable to that of T-mIL12-IRES (Fig. 2C).

Major point 5:

IL-12 expression is quantified by ELISA. However, data on transgene function, e.g. lymphocyte activation by virus-encoded IL-12, is lacking.

Response:

In order to address the reviewer’s comment, we performed an additional in vitro experiment to check the bioactivities of virus-encoded IL-12. IL-12 expressed by T-mfIL12 and T-mIL12-IRES both stimulated splenocytes, causing a release of interferon γ , and were thus confirmed bioactivity. The result is shown in newly added **Supplementary Figure 3**.

The following section was added to the Methods (p.13, 2nd paragraph):

Interferon- γ release assay. Conditioned media of Vero cells infected with T-mfIL12 or T-mIL12-IRES at MOI=1 were collected 48 h post-infection. VP-SFM medium (11681020, Thermo Fisher Scientific) was used for dilution. The spleen from A/J mice was harvested, and cell suspensions were prepared. Splenocytes were subjected to the conditioned media or recombinant mouse IL-12 (rIL12, 095-05331, Wako, Japan) for 48 h and interferon- γ (IFN γ) levels were measured in duplicates using mouse IFN γ Uncoated ELISA kit (88-7314, ThermoFisher Scientific).

The following sentence was added to the Results, “In vitro characteristics of T-mfIL12 and T-mIL12-IRES” section (p.18, 1st paragraph, last sentence):

IL-12 expressed by T-mfIL12 and T-mIL12-IRES both stimulated splenocytes, causing a release of IFN γ , and were thus confirmed bioactive (Supplementary Figure 3).

Major point 6:

The authors claim that T-mfIL-12 was significantly more efficacious than T-mIL12-IRES. However, this is based on selected datapoints, such as tumor volume on day 25 in the TRAMPC2 model. Overall there does not appear to be a meaningful difference.

Response:

We consider comparison of tumor volumes at later datapoints of subcutaneous tumor model experiments is an important statistical index, and have been performing it for our previous studies. However, to address the reviewer's concern, we additionally performed two-way ANOVA with Bonferroni's multiple comparisons test to analyze the significance between groups for subcutaneous tumor experiments, namely Figures 4A, 4B and newly added 4C.

Based on two-way ANOVA test, only T-mfIL12, but not T-mIL12-IRES, was significantly more efficacious than T-01 in the unilateral TRAMP-C2 subcutaneous tumor model (Fig. 4A). In the bilateral subcutaneous Neuro2a model (Fig. 4B), both T-mfIL12 and T-mIL12-IRES were significantly more efficacious than T-01 in the treated side as well as in the untreated side, and T-mfIL12 was more efficacious than T-mIL12-IRES in the untreated side.

Further, to allow comparison of the antitumor effects of T-mfIL12 and T-mIL12-IRES in detail, data on the tumor growth of individual animals for experiments of Figures 4A and 4B are provided as Supplementary Figure 4.

The following sentences are added to the Results, "In vivo efficacy of T-mfIL12 and T-mIL12-IRES section" (p.19, 2nd paragraph, p.20, 2nd paragraph, p.21, 1st paragraph):

Between groups, only T-mfIL12, but not T-mIL12-IRES, was significantly more efficacious than T-01 ($p=0.027$, two-way ANOVA; Fig. 4A).

Between groups, all three viruses were significantly more efficacious than mock in the treated side ($p=0.001$ for T-01 and $p<0.001$ for both T-mfIL12 and T-mIL12-IRES, two-way ANOVA) and T-mfIL12 and T-mIL12-IRES than mock in the untreated side ($p<0.001$ for both viruses, two-way ANOVA; Fig. 4B). Also, both T-mfIL12 and T-mIL12-IRES were significantly more efficacious than T-01 as group in the treated side ($p<0.001$ and $p=0.006$ for T-mfIL12 and T-mIL12-IRES, respectively, two-way ANOVA) as well as in the untreated side ($p<0.001$ and $p=0.011$ for T-mfIL12 and T-mIL12-IRES, respectively, two-way ANOVA; Fig. 4B).

Between groups, T-mfIL12 was more efficacious than T-mIL12-IRES in the untreated side ($p=0.018$, two-way ANOVA; Fig. 4B).

Major point 7:

In the Neuro2a model, the control virus has barely any effect. What is the benefit of virus-encoded IL-12 compared to direct intratumoral injection of IL-12? Potential advantages should be backed up with data.

Response:

To address the reviewer's question, we performed an additional *in vivo* experiment and compared the efficacy of T-mfIL12 with that of direct intratumoral injections with recombinant IL-12 in A/J mice harboring bilateral subcutaneous Neuro2a tumors. The results are shown in the newly added **Supplementary Figure 5**. When subcutaneous Neuro2a tumors were established, recombinant IL-12 (0.5 μ g), T-01 (5×10^4 pfu) without or with rIL-12, T-mfIL12 (5×10^4 pfu) or mock was inoculated into the left tumors only on days 0 and 4. In the treated side, rIL-12, T-01, T-01+rIL-12 and T-mfIL12 were all significantly more efficacious than mock ($p < 0.001$ vs mock for all, two-way ANOVA). In contrast, in the untreated side, rIL-12 alone showed no significant antitumor effect compared with mock, whereas T-01+rIL-12 and T-mfIL12 showed a significantly higher efficacy than mock ($p = 0.039$ and $p < 0.001$ vs mock, respectively, two-way ANOVA). Further, in the untreated side, T-mfIL12, but not T-01+rIL-12, was significantly more efficacious than rIL-12 ($p = 0.003$, two-way ANOVA). The results implicate that the potential advantage of virus-encoded IL-12 is that it acts more effectively than recombinant IL-12 on systemic antitumor immunity.

The following paragraph is newly added to the Results, "In vivo efficacy of T-mfIL12 and T-mIL12-IRES" section (p.23, 2nd paragraph):

To evaluate the benefit of expressing IL-12 as an armed oncolytic HSV-1, we compared the efficacy of T-mfIL12 with that of direct intratumoral injections with recombinant IL-12 in A/J mice harboring bilateral subcutaneous Neuro2a tumors (Supplementary Figure 5). When established tumors reached approximately 5 mm in diameter, rIL-12 (0.5 μ g)³⁴, T-01 (5×10^4 pfu) without or with rIL-12, T-mfIL12 (5×10^4 pfu) or mock was inoculated into the left tumors only on days 0 and 4. In the treated side, rIL-12, T-01, T-01+rIL-12 and T-mfIL12 were all significantly more efficacious than mock ($p < 0.001$ vs mock for all, two-way ANOVA). In contrast, in the untreated side, rIL-12 alone showed no significant antitumor effect compared with mock, whereas T-01+rIL-12 and T-mfIL12 showed a significantly higher efficacy than mock ($p = 0.039$ and $p < 0.001$ vs mock, respectively, two-way ANOVA). Further, in the untreated side, T-mfIL12, but not T-01+rIL-12, was significantly more efficacious than rIL-12 ($p = 0.003$, two-way ANOVA). The results implicate that virus-encoded IL-12 acts more effectively than recombinant IL-12 on systemic antitumor immunity.

34 Caminschi, I., et al. Interleukin-12 induces an effective antitumor response in malignant mesothelioma. *Am. J. Respir. Cell Mol. Biol.* **19**, 738-746 (1998).

Major point 8:

No data on transgene expression and target engagement in vivo is presented. What intratumoral levels of IL-12 are reached after treatment with the respective virus variants?

What effects does treatment have on anti-tumor immunity? The bilateral flank model implies there may be an abscopal effect. Can tumor-specific immune responses be detected, e.g. by ELISpot?

Response:

To address the reviewer's questions, we performed additional in vivo experiments.

First, to assess the in vivo transgene expression and function by T-mfIL12 and T-mIL12-IRES, we measured the in vivo IL-12 and IFN γ levels in the unilateral subcutaneous Neuro2a model. The results are shown in newly added **Figure 5B**. Established tumors were inoculated with T-01, T-mfIL12, T-mIL12-IRES (2×10^6 pfu) or mock, and sera and tumor samples were collected from three mice per group on days 1, 3 and 6. In the tumor, the level of IL-12 peaked on day 1 and gradually decreased by days 3 and 6 for both T-mfIL12 and T-mIL12-IRES (Fig. 5B). The levels of intratumoral IL-12 of T-mfIL12 were significantly higher than those of T-mIL12-IRES at all time points ($p=0.018$, $p=0.016$ and $p=0.046$ for days 1, 3 and 6, respectively, one-way ANOVA). Much lower levels of IL-12 were detected from the serum, but the serum IL-12 level for T-mfIL12 was still higher than that for T-mIL12-IRES on day 1 ($p=0.047$, one-way ANOVA). IFN γ was detected in the tumor from day 1 for all three viruses, which increased by day 3 (Fig. 5B). IFN γ was detected from the serum with IL-12-expressing viruses only, which peaked on day 1. The IFN γ levels of T-mfIL12 were significantly higher than those of T-mIL12-IRES both in the tumor and serum on day 1 ($p=0.014$ and $p=0.027$, tumor and serum, respectively, one-way ANOVA).

Secondly, to assess the specific antitumor immune responses elicited by T-mfIL12 and T-mIL12-IRES, we performed ELISpot assays in the unilateral subcutaneous Neuro2a model. The results of IFN γ -releasing splenocytes responding to Neuro2a cell stimulation are shown in newly added **Figure 5C**. ELISpot assay results including control Sal/N cell stimulation are shown in **Supplementary Figure 6**. Established tumors were inoculated with T-01, T-mfIL12, T-mIL12-IRES (5×10^4 pfu) or mock on days 0 and 3, and the spleen was harvested on day 6. By ELISpot assay, splenocytes from T-mfIL12-treated mice showed a significantly higher number of IFN γ release stimulated by Neuro2a cells than those from T-01- and T-mIL12-IRES- treated ones ($p=0.005$ and $p=0.004$ vs T-01 and T-mIL12-IRES, respectively, one-way ANOVA; Fig. 5C). Such responses

were not observed with SaI/N cells, control cells derived from A/J mouse strain (Supplementary Figure 6). No significant difference in number of IL-4 releasing splenocytes was observed among the three virus-treated groups (Fig. 5C).

“*In vivo* IL-12 and IFN γ levels” and “Enzyme-linked immunospot (ELISpot) assay” sections are newly added to the Methods (p.15, 2nd and 3rd paragraph).

Two new sections describing the above results are added to the Results (p.23, 3rd and p.24, 2nd paragraph):

In vivo levels of IL-12 and IFN γ by T-mfIL12 and T-mIL12-IRES

To assess the *in vivo* transgene expression and function by T-mfIL12 and T-mIL12-IRES, we measured the *in vivo* IL-12 and IFN γ levels in the unilateral subcutaneous Neuro2a model. Established tumors were inoculated with T-01, T-mfIL12, T-mIL12-IRES (2×10^6 pfu) or mock, and sera and tumor samples were collected on days 1, 3 and 6 (n=3 per group). In the tumor, the level of IL-12 peaked on day 1 and gradually decreased by days 3 and 6 for both T-mfIL12 and T-mIL12-IRES (Fig. 5B). The levels of intratumoral IL-12 for T-mfIL12 were significantly higher than those for T-mIL12-IRES at all time points ($p=0.018$, $p=0.016$ and $p=0.046$ for days 1, 3 and 6, respectively, one-way ANOVA). The levels of IL-12 detected from the serum were remarkably lower (approximately 2log lower) than those in the tumor. Correlating with the intratumoral IL-12, the serum IL-12 level for T-mfIL12 was higher than that for T-mIL12-IRES on day 1 ($p=0.047$, one-way ANOVA). IFN γ was detected in the tumor from day 1 for all three viruses, which increased by day 3 (Fig. 5B). IFN γ was detected from the serum with IL-12-expressing viruses only, which peaked on day 1. The IFN γ levels for T-mfIL12 were significantly higher than those for T-mIL12-IRES both in the tumor and serum on day 1 ($p=0.014$ and $p=0.027$, tumor and serum, respectively, one-way ANOVA).

Specific antitumor immune responses by T-mfIL12 and T-mIL12-IRES

To assess the specific antitumor immune responses elicited by T-mfIL12 and T-mIL12-IRES, we performed ELISpot assays in the unilateral subcutaneous Neuro2a model. Established tumors were inoculated with T-01, T-mfIL12, T-mIL12-IRES (5×10^4 pfu) or mock on days 0 and 3, and the spleen was harvested on day 6. By ELISpot assay, splenocytes from T-mfIL12-treated mice showed a significantly higher number of IFN γ release stimulated by Neuro2a cells than those from T-01- and T-mIL12-IRES- treated ones ($p=0.005$ and $p=0.004$ vs T-01 and T-mIL12-IRES, respectively, one-way ANOVA; Fig. 5C). Such responses were not observed with SaI/N cells, control cells derived from A/J mouse strain (Supplementary Figure 6). No significant difference in number of IL-4 releasing splenocytes was observed among the three virus-treated

groups (Fig. 5C).

Major point 9:

Do the IL-12 encoding viruses promote tumor infiltration with immune cells or activation of tumor-infiltrating lymphocytes? Data on such mechanisms of action is needed to assess the therapeutic potential of the novel viruses.

Response:

To answer the reviewer's question, we performed an additional in vivo experiment to assess whether IL-12-expressing viruses promote tumor-infiltrating lymphocytes by immunohistochemistry. The results are shown in newly added **Figure 5A**. Bilateral subcutaneous Neuro2a tumors were generated in A/J mice, left tumors only were inoculated with T-mfIL12, T-mIL12-IRES, T-01 (2×10^5 pfu) or mock on days 0 and 3, and the tumors were harvested on day 6. An increased infiltration of CD4⁺ and CD8⁺ lymphocytes were observed in the tumor for all three viruses, both in the treated and the untreated side, which was apparently most prominent with T-mfIL12 (Fig. 5A). HSV-1 positive cells were observed in the tumor with all viruses in the treated side, but none of the viruses caused HSV-1 positivity in the untreated side (Fig. 5A).

“Immunohistochemistry” section is newly added to the Methods (p.14, 2nd paragraph).

The following paragraph describing the above results is added to the Results, “In vivo efficacy of T-mfIL12 and T-mIL12-IRES” section (p.22, 2nd paragraph):

We further assessed by immunohistochemistry whether IL-12-expressing viruses promote infiltration of lymphocytes into the tumor. Bilateral subcutaneous Neuro2a tumors were generated in A/J mice, left tumors only were inoculated with T-mfIL12, T-mIL12-IRES, T-01 (2×10^5 pfu) or mock on days 0 and 3, and the tumors were harvested on day 6. An increased infiltration of CD4⁺ and CD8⁺ lymphocytes were observed in the tumor for all three viruses, both in the treated and the untreated side, most prominently with T-mfIL12 (Fig. 5A). HSV-1 positive cells were observed in the tumor with all viruses in the treated side, but none of the viruses caused HSV-1 positivity in the untreated side (Fig. 5A).

Following statement is added in the Discussion (p.26, 1st paragraph):

A significantly higher amounts of IL-12 are detected in T-mfIL12-treated tumors than T-mIL12-IRES-treated ones, leading to stronger antitumor immune responses as evidenced by IFN γ measurement, ELISpot assay and immunohistochemistry in the poorly immunogenic, Neuro2a tumor model in HSV-1 sensitive A/J mice.

Major point 10:

What are systemic levels of IL-12 after treatment with the virus variants? This is an important safety aspect with respect to clinical application.

Response:

As described above in our response to the Major point 8 of the reviewer, we measured the serum IL-12 levels 1, 3 and 6 days after inoculating subcutaneous Neuro2a tumors with T-01, T-mfIL12, T-mIL12-IRES (2×10^6 pfu) or mock. The results are shown in Figure 5B. The levels of IL-12 detected from the serum were remarkably lower (approximately 2log lower) than those in the tumor.

Comment 10:

In the bilateral model, was spread of the virus to the non-treated tumor excluded?

Response 10:

As described in our response to the Major point 9 of the reviewer, we performed an in vivo experiment and assessed the presence of HSV-1 by immunohistochemistry. The results are shown in newly added **Figure 5A**. Bilateral subcutaneous Neuro2a tumors were generated in A/J mice, left tumors only were inoculated with T-mfIL12, T-mIL12-IRES, T-01 (2×10^5 pfu) or mock on days 0 and 3, and the tumors were harvested on day 6. HSV-1 positive cells were observed in the tumor with all viruses in the treated side, but HSV-1 positivity was not detected in the untreated side (Fig. 5A).

Also, we have previously reported that, when an oncolytic HSV-1 is inoculated into subcutaneous N18 tumors in A/J mice, the virus was not detected by PCR in remote tumors (1).

1. Todo T, Rabkin SD, Sundaresan P, Wu A, Meehan KR, Herscowitz HB, Martuza RL: Systemic antitumor immunity in experimental brain tumor therapy using a multimitated, replication-competent herpes simplex virus. Hum Gene Ther 10: 2741-2755, 1999.

Major comment 12:

All data presented comes from mouse cell lines and mouse models. At the end of the discussion, it is stated that a phase 1/2 clinical trial with T-hIL-12 is ongoing. It would be very interesting to see data on the human IL-12 variant virus such as replication and cytotoxicity in human tumor cells, IL-

IL-12 expression levels, IL-12 function.

Response:

We appreciate the reviewer's comments, but this paper focuses on evaluating the optimal methods for expressing IL-12 as a payload for G47Δ-based oncolytic HSV-1 by comparing different types of IL-12 transgene. Although we have preclinical data of T-hIL12, a G47Δ-based oncolytic HSV-1 that expresses human IL-12 as a functional fusion peptide, that led the ongoing clinical trial, inclusion of such data is beyond the objective of this paper and will be described elsewhere.

Major comment 13:

For the in vivo data, it would be helpful to show individual tumor growth curves for all animals, also beyond day 30/ day 15.

Response:

As described in our response to the Major point 6 of the reviewer, data on the tumor growth of individual animals for experiments of Figures 4A and 4B are provided as Supplementary Figure 4. In compliance with the recommendation by the Institutional Animal Care and Use Committee, animals were sacrificed when the maximum diameter of a subcutaneous tumor reached 22 mm, therefore we don't have data beyond day 30/ day 15.

Minor comment 1:

Line 183: Why was temperature shifted to 39.5°C for the ELISA sampling? Please explain.

Response:

HSV-1 wild-type strain F, the parental strain of G47Δ as well as T-01, T-mfIL12 and T-mIL12-IRES, is temperature sensitive⁵⁽²⁾, therefore does not replicate at 39.5°C. For ELISA sampling, cells infected with the viruses were incubated at 39.5°C to prevent early cell destruction due to viral replication. This is now briefly explained in the Methods (p.12, 2nd paragraph).

All viruses used in this study derive from HSV-1 strain F, are therefore temperature sensitive, and do not replicate at 39.5°C⁵.

2. Knipe, D.M., et al. Molecular genetics of herpes simplex virus. VI. Characterization of a temperature-sensitive mutant defective in the expression of all early viral gene products. *J. Virol.* **38**, 539-547 (1981).

Minor comment 2:

Lines 220 - 221: Instead of "front" and "back", terms such as "upstream" and "downstream" should be used to describe genome sequences.

Response:

In compliance with the reviewer's comments, we changed the terms to "upstream" and "downstream" instead of "front" and "back" in the Results, "Construction of T-mfIL12 and T-mIL12-IRES" section (p.17, 1st paragraph).

Minor comment 3:

Figure 1: Perhaps indicate hours post infection instead of days.

Response:

In Figure 3, we changed the labels of horizontal axis to 'hours post infection' instead of 'days'.

Responses to Reviewer #2

General Comments:

Fukuhara et al. report on the modification of G47 Δ which is a triple-mutated oncolytic herpes simplex virus type 1 (HSV-1) via the expression of murine IL-12. The authors explored two modalities to accomplish this (i.e., via the creation of T-mfIL12 which expresses murine IL-12 as a fusion peptide, with the genes of two subunits [p35 and p40] linked by bovine elastin motifs, and via the creation of T-mIL12-IRES which co-expresses the subunits, with the two genes separated by an internal ribosome entry site [IRES] sequence).

In so doing, Fukuhara et al examined the expression of functional IL-12 via ELISA, replication of parent viruses and ultimately in vitro/in vivo cytotoxicity/efficacy (using a neuroblastoma and prostate models).

While this is a solid contribution to the field, the role of IL-12 in oHSVs has been well defined and the findings while very important may be more incremental in nature. A few pertinent suggestions for the authors are highlighted below as they consider revising/resubmission of the manuscript.

Response:

We thank the reviewer for the general comments. Point by point responses to the reviewer's suggestions are listed below.

Major point 1:

The authors continue to report that G47delta was successful in a Phase 2 trial; aside from an abstract published in 2019 (PMCID: PMC6847570) this reviewer is unaware of a formal publication reporting on the trial. I would suggest the authors discuss this and/or provide the Japanese clinical trial number in future versions of the manuscript.

Response:

In compliance with the reviewer's comments, we now provide the clinical trial registration numbers.

The following sentences are modified to include the clinical trial registration numbers in the Introduction (p.5, 1st paragraph):

The safety of G47Δ has been tested in several clinical trials in Japan, including those for glioblastoma (UMIN00002661)¹⁷, castration resistant prostate cancer (UMIN000010463), metastatic prostate cancer (jRCTs033210603), olfactory neuroblastoma (UMIN000011636, jRCTs033180325), and malignant pleural mesothelioma (UMIN000034063, jRCTs033180326). Recently, the phase II trial (UMIN000015995) for glioblastoma led to the approval of G47Δ as a new drug for malignant glioma in Japan¹⁸.

Also, we recently published the results of the Phase II clinical trial and the phase I/II (first-in-human) clinical trial of G47Δ in patients with glioblastoma. These two papers are now cited in the Introduction and included in the Reference.

17 Todo, T., et al. A phase I/II study of triple-mutated oncolytic herpes virus G47Δ in patients with progressive glioblastoma *Nat Commun* **13**: 4119 (2022).

18 Todo, T., et al. Intratumoral oncolytic herpes virus G47Δ for residual or recurrent glioblastoma: a phase 2 trial. *Nat Med* **28**: 1630-1639 (2022).

Major point 2:

If the overall guise of this paper is to discuss/explore the preclinical rationale for a human IL-12 fusion protein prior to beginning clinical trials in malignant melanoma as referenced, why employ the preclinical models chosen (i.e., neuroblastoma and prostate) as opposed to melanoma? Further

the folding of the human version of the fusion protein should be explored given the distinct differences between human and murine versions of IL-12 protein.

Response:

We appreciate the reviewer's comments, but this paper focuses on evaluating the optimal methods for expressing IL-12 as a payload for G47 Δ -based oncolytic HSV-1 by comparing different types of IL-12 transgene, not restricting the cancer type. We consider the poorly immunogenic, Neuro2a tumor model in HSV-1 sensitive A/J mice, the main tumor model used in this study, to be one of the most suitable models to evaluate the functions of immunostimulatory payloads of oncolytic HSV-1. The results of this study led to create a G47 Δ -based oncolytic HSV-1 that expresses human IL-12 as a functional fusion peptide (T-hIL12). The functions of human IL-12 expressed by T-hIL12, including the folding, have been studied before starting the ongoing clinical trial, but inclusion of such data is beyond the objective of this paper and will be described elsewhere. Therefore, we added a simple statement in the Discussion as follows (p.29. last sentence):

T-hIL12 has been well characterized, including the functions of expressed human IL-12, and T-hIL12 is currently used in ongoing, investigator-initiated, phase 1/2 clinical trial in patients with malignant melanoma in Japan (jRCT2033190086).

We also added the following sentence in the Discussion explaining why we chose the Neuro2a model (p.26, 1st paragraph, last sentence):

This tumor model was used in this study, because we consider it one of the most suitable models to evaluate the functions of immunomodulatory payloads of oncolytic HSV-1.

Major point 3:

The decreased viral yields noted in the IL-12 variants vs the parent G47 Δ virus should be explored in detail; next generation sequencing may be needed to confirm genomic stability given the insertions (w/ the understanding that GC rich regions within α HSVs may be difficult to cover).

Response:

As the reviewer pointed out, a mild decrease in viral yields was observed in the IL-12-expressing viruses compared with G47 Δ . The purpose of this paper, however, is not the comparison between G47 Δ and the IL-12 variants, but the comparison between different types of IL-12 transgene as payloads of oncolytic HSV-1. As mentioned above, G47 Δ -based oncolytic HSV-1 expressing fusion-type human IL-12 (T-hIL12) has been well characterized before the start of the ongoing clinical trial for malignant melanoma, and is now stated in the last sentence of the Discussion.

Major point 4:

Efficacy of survival in a xenograft model should also be explored removing the potential confounder of IL-12/adaptive immunity. This would ensure that replication is equivalent in vivo given the novel transgene insertions.

Response:

In compliance with the reviewer's comment, we performed an additional in vivo experiment using NOG mice. To confirm that the enhanced efficacy of T-mfIL12 and T-mIL12-IRES over T-01 (control virus) is due to the immune actions of virus-encoded IL-12, the efficacy was evaluated in NOG mice, eliminating the effects of adaptive immunity as well as NK activity. When established bilateral subcutaneous Neuro2a tumors reached approximately 5 mm in diameter, T-01, T-mfIL12, T-mIL12-IRES (1×10^6 pfu) or mock was inoculated into the left tumors only on days 0 and 3. In the treated side, all three viruses exhibited significant efficacy compared with mock ($p < 0.001$ vs mock for all viruses, two-way ANOVA). However, the effect of IL-12 expression was completely abolished in NOG mice for both T-mfIL12 and T-mIL12-IRES, and there was no difference in efficacy among the three viruses T-01, T-mfIL12 and T-mIL12-IRES. All three viruses had no effect on the contralateral untreated tumors. The results indicate that the enhanced efficacy by T-mfIL12 and T-mIL12-IRES in immunocompetent mouse models is indeed via the immune actions of virus-encoded IL-12.

The results are shown in newly added Figure 4C.

The use of NOG mice is now mentioned in the Methods, "Animal experiments section" (p.13, 3rd paragraph):

The following experimental results are added as a new paragraph in the Results, "In vivo efficacy of T-mfIL12 and T-mIL12-IRES section" (p.21, 2nd paragraph):

To confirm that the enhanced efficacy of T-mfIL12 and T-mIL12-IRES over T-01 is due to the immune actions of virus-encoded IL-12, the efficacy was further evaluated in NOG mice, eliminating the effects of adaptive immunity as well as NK activity (Fig. 4C). When established bilateral subcutaneous Neuro2a tumors reached approximately 5 mm in diameter, T-01, T-mfIL12, T-mIL12-IRES (1×10^6 pfu) or mock was inoculated into the left tumors only on days 0 and 3. In the treated side, all three viruses exhibited significant efficacy compared with mock ($p < 0.001$ vs mock for all viruses, two-way ANOVA). However, the effect of IL-12 expression was completely

abolished in NOG mice for both T-mfIL12 and T-mIL12-IRES, and there was no difference in efficacy among the three viruses T-01, T-mfIL12 and T-mIL12-IRES. All three viruses had no effect on the contralateral untreated tumors (Fig. 4C).

The following statement is added in the Discussion (p.26, 1st paragraph):

The function of virus-encoded IL-12 is confirmed to be immune-mediated, as the enhanced efficacy is abolished in NOG mice.

Minor comment 1:

There are a litany of grammatical errors throughout the manuscript that should be addressed; style and flow should also be improved throughout the paper (this may be accomplished via engagement of a native English speaker).

Response:

The revised manuscript has been checked by a native English speaker.

Minor comment 2:

Figure 3 is referenced in Results prior to Figure 2; the authors should therefore realign the flow of the figures.

Response:

All figures are now numbered in the order of citation.

Minor comment 3:

The authors should explore expression of p35 vs p40 in the IRES construct to ensure 1:1 levels of transcription/translation; P2A and T2A domains might also be considered as opposed to IRES moving forward.

Response:

As the reviewer points, IRES might cause an uneven expression of the two subunit genes, leading to an overexpression of the p40 gene closer to the promoter. An excess free p40 subunit might inhibit the activity of IL-12 in murine models²⁹. These concerns are already stated in the Introduction (p. 7, 1st paragraph, last sentences).

We checked the expressions of the p35 and p40 subunits by T-mfIL12 and T-mIL12-IRES

by performing ELISA for mouse IL-12 p35 and mouse IL-12 p40 using conditioned media collected from Vero cells infected with T-mfIL12 or T-mIL12-IRES. When we first used recombinant mouse IL-12 (2 µg/ml) as a control which should show 1:1 ratio for p35 and p40, we detected 709µg/ml of p35 and 113µg/ml of p40, resulting in 6.3:1 ratio. This discrepancy was presumably due to difference in affinity of antibodies used for the p35 and p40 units. Therefore, we adjusted the values based on this control result, and the adjusted ratio of p35 and p40 expressed by T-mfIL12 and T-mIL12-IRES were 1.8:1 and 10:1, respectively. Accordingly, the following sentence is added to the Results (p.18, 1st paragraph):

The adjusted expression ratios of p35 and p40 subunits were 1.8:1 and 10:1 for T-mfIL12 and T-mIL12-IRES, respectively.

We also added descriptions in the Methods, “In vitro *IL-12* expression measurement” section as follows (p.13, 1st paragraph):

For the detection of the p35 subunit and p40 subunit separately, ELISA kit for mouse IL12A (IL12 p35) (SEA059Mu, Cloud-Clone, TX) and ELISA kit for mouse IL12B (IL12 p40) (SEA058Mu, Cloud-Clone, TX) were used. The expression ratios of p35 and p40 subunits were adjusted based on control (recombinant mouse IL-12, 095-05331, Wako, Japan).

We agree with the reviewer that the use P2A and T2A domains between the p35 gene and p40 gene would be a good option.

Minor comment 4:

The M032 clinical trial (<https://clinicaltrials.gov/ct2/show/NCT02062827>) should be referenced within the discussion.

Response:

In compliance with the reviewer’s comment, we now refer to the M032 clinical trial in the Discussion (p.25, 1st paragraph):

M032 is used in a phase 1 study in patients with recurrent malignant glioma (NCT02062827).

Minor comment 5:

There have also been published clinical trials of IL12 gene therapy for cancer in the last 3-4 years using adenoviral vectors. These should also be cited since it is relevant to the paper.

Response:

In compliance with the reviewer's request, we add the following statement in the Discussion citing additional references (p.27, 1st paragraph):

Several clinical trials of IL-12 gene therapy have been performed lately using adenovirus vectors^{42,43}.

42. Chiocca, E.A., et al. Regulatable interleukin-12 gene therapy in patients with recurrent high-grade glioma: Results of a phase 1 trial. *Sci. Transl. Med.* **11(505)**, eaaw5680 (2019).

43. Chiocca, E.A., et al. Combined immunotherapy with controlled interleukin-12 gene therapy and immune checkpoint blockade in recurrent glioblastoma: An open-label, multi-institutional phase I trial. *Neuro. Oncol.* **24**, 951-963 (2022).

Minor comment 6:

Can the authors comment on the immunogenicity of the bovine linker? Do they anticipate reactions in patients given repeat dosing/exposures?

Response:

Theoretically, the bovine elastin motifs might act immunogenic in patients with repeated dosing. G47Δ and other G47Δ-derived viruses also express the lacZ gene (β-galactosidase), another potentially immunogenic protein, but in the clinical trials with G47Δ, we have not experienced unwanted reactions that may have been caused by β-galactosidase. Therefore, we think reactions to the bovine elastin motifs should not be much anticipated.

Responses to Reviewer #3

General comments

In this paper they have developed a triple-mutated oncolytic HSV-1 armed with human IL-12. The oncolytic virus they have chosen to work with is currently used in phase 1/2 trial for malignant melanoma. They have created two viruses using two approaches to express IL-12: T-mfIL12 expresses murine IL-12 as a fusion peptide, with the genes of two subunits (p35 and p40) linked by bovine elastin motifs, and T-mIL12-IRES co-expresses the subunits, with the two genes separated by

an internal ribosome entry site (IRES) sequence. Their claim is the fusion-type expression of IL-12 is superior to co-expression of separate subunits, and results in higher production of functional IL-12 molecules.

Response:

We thank the reviewer for the general comments. Point by point responses to the reviewer's specific comments are listed below.

Comment 1:

The use of OV's expressing IL-12 is exciting as this overcomes the systemic toxicity of IL-12 given as a cytokine therapy. The generation of IL-12 armed HSV as well as other viruses has been attempted and published in the past. For example, Haghghi et al, inserted IL-12 in the ICP34.5 deleted HSV-1 backbone to improve the oncolytic properties and provide an immune-stimulatory effects in colorectal cancer cells. <https://doi.org/10.1016/j.micpath.2021.105164>. They used pCDNA3.1+ CMV-hIL12 containing the p40 and p35 subunits of human IL-12. More recent studies using oncolytic HSV armed with IL-12 were shown to have vaccine like properties doi:10.1371/journal.ppat.1007209.

Response:

We thank the reviewer for the additional information regarding oncolytic HSV-1 armed with IL-12. The following statement is added to the Discussion citing additional references (p.28, 1st paragraph):
Other approaches to arm oncolytic HSV-1 with IL-12 have been reported, including insertion of the human IL-12 gene in the ICP47- and ICP34.5-deleted HSV-1 backbone ($\Delta 47/\Delta 34.5/IL12$)⁵² and arming fully-virulent, HER2-retargeted oncolytic HSV-1 with murine IL-12 (R-115)^{53,54}.

52. Haghghi-Najafabadi, N., et al. Oncolytic herpes simplex virus type-1 expressing IL-12 efficiently replicates and kills human colorectal cancer cells. *Microb. Pathog.* **160**, 105164 (2021).

53. Menotti, L., et al. HSV as A Platform for the Generation of Retargeted, Armed, and Reporter-Expressing Oncolytic Viruses. *Viruses* **10**, 352 (2018).

54. Lioni, V., et al. A fully-virulent retargeted oncolytic HSV armed with IL-12 elicits local immunity and vaccine therapy towards distant tumors. *PLoS Pathog.* **14**, e1007209 (2018).

Comment 2:

There is also some confusion regarding the use of this virus in clinical trials, G47 Δ was approved as a new drug for malignant glioma in Japan but the choice of wording in the abstract and lay summary

suggests the IL-12 armed virus is currently in clinical trials for melanoma. Where is this trial?
Are they recruiting?

Response:

T-hIL12 is a G47 Δ -based oncolytic HSV-1 that expresses human IL-12 as a functional fusion peptide, different from G47 Δ . As mentioned in the Discussion, T-hIL12 is currently used in investigator-initiated, phase 1/2 clinical trial in patients with malignant melanoma in Japan (jRCT2033190086), and the trial is recruiting.

To clarify, we added the word “ongoing” in the last sentences of the Plain language summary and the Discussion:

(p. 4) This study led to the creation of triple-mutated oncolytic HSV-1 armed with human IL-12 currently used in **ongoing** phase 1/2 trial for malignant melanoma.

(p. 29) T-hIL12 is currently used in **ongoing**, investigator-initiated, phase 1/2 clinical trial in patients with malignant melanoma in Japan (jRCT2033190086).

Comment 3:

More clarity regarding the choice of the intratumoural dose and why 2 doses of HSV were given is needed.

Response:

A/J mice and A/J-derived Neuro2a cells are susceptible to HSV-1 infection whereas C57BL/6 mice and C57BL/6-derived TRAMP-C2 cells are relatively resistant³³. The doses were chosen based on our experience with these models. Two doses are given to minimize unintended deviation caused by intratumoral injection techniques. The second dose is given 3 days after the first to avoid deaths by anesthesia.

33. Lopez, C. Genetics of natural resistance to herpesvirus infections in mice. *Nature* **258**, 152-153 (1975).

To clarify the reason for the choice of dose, the following descriptions are added to the Results:

(p.19, 2nd paragraph). **C57BL/6 mice and C57BL/6-derived TRAMP-C2 cells are relatively resistant to HSV-1 infection³³, and the dose (5×10^6 pfu) was determined based on our previous studies⁹.**

(p.20, 2nd paragraph). A/J mice and A/J-derived Neuro2a cells are susceptible to HSV-1 infection³³, and the dose (5×10^4 pfu) was determined based on our experience with this model²⁰.

Comment 3:

Also why the tumours were grown to be so large.

Response:

In the original manuscript, we expressed the tumor volume according to the formula, length \times width \times height, therefore the tumors were seemingly grown to be large, where in fact they were not. We changed the formula for the tumor volume to length \times width \times height \times 0.52, so the tumor volumes are now close to the actual sizes.

The following sentence is modified in the Methods, “Animal experiments” section (p.14, 1st paragraph):

Tumor growth was determined by measuring tumor volume (length \times width \times height \times 0.52) twice a week.

Comment 4:

The paper also felt short in the analysis of the mice. It was disappointing that no immune analysis was carried out to determine how the IL-12 viruses changed the tumour microenvironment especially as this was in fully immunocompetent mice.

Response:

As we respond to the Major point 8 of Reviewer 1, we performed several additional in vivo experiments to show immune responses by IL-12 expressing viruses.

To assess the specific antitumor immune responses elicited by T-mfIL12 and T-mIL12-IRES, we performed ELISpot assays in the in the unilateral subcutaneous Neuro2a model. The results of IFN γ -releasing splenocytes responding to Neuro2a cell stimulation are shown in newly added **Figure 5C**. ELISpot assay results including control Sal/N cell stimulation are shown in **Supplementary Figure 6**. Established tumors were inoculated with T-01, T-mfIL12, T-mIL12-IRES (5×10^4 pfu) or mock on days 0 and 3, and the spleen was harvested on day 6. By ELISpot assay, splenocytes from T-mfIL12-treated mice showed a significantly higher number of IFN γ release stimulated by Neuro2a cells than those from T-01- and T-mIL12-IRES- treated ones ($p=0.005$ and $p=0.004$ vs T-01 and T-mIL12-IRES, respectively, one-way ANOVA; Fig. 5C). Such responses

were not observed with SaI/N cells, control cells derived from A/J mouse strain (Supplementary Figure 6). No significant difference in number of IL-4 releasing splenocytes was observed among the three virus-treated groups (Fig. 5C).

As we respond to Major point 9 of Reviewer 1, we further performed an in vivo experiment to assess whether IL-12-expressing viruses promote tumor-infiltrating lymphocytes by immunohistochemistry. The results are shown in newly added **Figure 5A**. Bilateral subcutaneous Neuro2a tumors were generated in A/J mice, left tumors only were inoculated with T-mIL12, T-mIL12-IRES, T-01 (2×10^5 pfu) or mock on days 0 and 3, and the tumors were harvested on day 6. An increased infiltration of CD4⁺ and CD8⁺ lymphocytes were observed in the tumor for all three viruses, both in the treated and the untreated side, which was apparently most prominent with T-mIL12 (Fig. 5A). HSV-1 positive cells were observed in the tumor with all viruses in the treated side, but none of the viruses caused HSV-1 positivity in the untreated side (Fig. 5A).

Comment 5:

Was there an increase in IL-12 secretion in cell cultures following infection or in the mouse tumours following injection. This may have yielded more exciting data and determined if the methods of virus production produced effective viruses.

Response:

Comment 6:

Statistical analysis is appropriate but should have been described in the figure legends.

Response:

All statistical analyses are now described in the figure legends.

Reviewers' comments:

Reviewer #1 (Remarks to the Author):

The authors have addressed many of the comments and performed additional experiments to improve the manuscript.

I have a few remaining comments:

- Statistics: If three groups are in an experiment, ANOVA should be used, not t-test.
- What is the rationale for the choice of IL-12 dose (0.5 µg, Supplementary Figure 5) for intratumoral injection?
The dose should be aligned with the expected intratumoral levels of IL-12 after virus treatment.
- The immunohistochemistry data (Figure 5A) should be quantified.
- It seems the intratumoral levels of IL-12 are highest on the first day after treatment (Figure 5B). Is this from input virus? Non-purified virus suspensions can contain considerable amounts of transgene product.
This would not be transferable to an actual clinical treatment setting with highly purified/GMP-grade virus.

Figure 2D (in vitro) only includes one timepoint, no kinetics or the same timepoints as the in vivo data, so these data cannot be compared

and it is unclear which levels of IL-12 are actually reached by virus replication.

The authors state that "The temperature was shifted to 39.5°C to prevent viral replication for ELISA sampling."

Then the ELISA cannot measure the levels that are reached during viral replication.

So the same question applies to the in vitro data: Is the IL-12 that was detected from input virus?

Also, I would not use the term "conditioned media" but rather "supernatants" for these samples.

- The authors have changed how they calculate tumor volume. Was this not pre-specified in the animal protocol?

They now state "Tumor growth was determined by measuring tumor volume (length × width × height × 0.52) twice a week" - why 0.52?

Reviewer #2 (Remarks to the Author):

The authors have satisfactorily answered all my concerns

Reviewer #3 (Remarks to the Author):

The authors have taken time to take on board the comments and suggestions made by the

reviewers. The manuscript has been significantly improved and most of the reviewers comments have been addressed with the inclusion of new data and necessary corrections. The revised version is worthy of publication.

November 29, 2022

RE: COMMSMED-22-0076A Revised

Fusion peptide is superior to co-expressing subunits for arming oncolytic herpes virus with interleukin 12

by Hiroshi Fukuhara, Yuzuri Tsurumaki Sato, Jiangang Hou, Miwako Iwai and Tomoki Todo

We thank the Reviewers for re-reviewing our manuscript. Detailed responses to the comments of Reviewers are listed below. Revised sentences and words are shown in red.

Responses to Reviewer #1

General comments:

The authors have addressed many of the comments and performed additional experiments to improve the manuscript.

I have a few remaining comments:

Comment 1:

Statistics: If three groups are in an experiment, ANOVA should be used, not t-test.

Response:

In the first revision, we did use ANOVA for all experiments with three groups or more wherever ANOVA was applicable and appropriate, namely Figures 4A, 4B, 4C, 5B, 5C and former Supplementary Figure 5 (now Figure 6A). ANOVA was not used for Supplementary Figure 3, because “n” was 2 per group, so ANOVA was not applicable. However, we consider comparison of tumor volumes at later datapoints of subcutaneous tumor model experiments using t-test is also an important statistical index and have been performing it for our previous studies (as indicated in our response to Major point 6 of Reviewer #1 in our first revision). We consider performing both ANOVA and t-test in the same experiment (Figures 4A and 4B) rather provides more information. We also point that, in the review comments for our original manuscript, Reviewer #3 commented “Statistical analysis is appropriate” (the last comment). We think the current statistical analyses are performed and indicated appropriately, and ANOVA is already performed for those appropriate.

Comment 2:

What is the rationale for the choice of IL-12 dose (0.5 µg, Supplementary Figure 5) for intratumoral injection? The dose should be aligned with the expected intratumoral levels of IL-12 after virus treatment.

Response:

In our first revision, to perform an additional *in vivo* experiment using recombinant IL-12 to generate former Supplementary Figure 5 (now Figure 6A), we searched literatures for the use of recombinant IL-12 in mouse tumor models. There were not many publications that reported injection of recombinant IL-12 into mouse tumors, but cited reference (Ref. 34) reported on intratumoral injections of recombinant IL-12 that caused efficacy. The report used 0.5 µg (500 ng) for intratumoral injection, so the same dose was chosen for our additional experiment in the first revision. The intratumoral levels of expressed IL-12 by IL-12-encoded viruses were not known until we newly performed a separate *in vivo* experiment (Figure 5B), so there were no data on “expected intratumoral levels” at the time of the first revision. However, we now know from Figure 5B that the highest intratumoral IL-12 level by intratumoral injection with 2×10^6 pfu of T-mfIL12 was calculated to be 45.7 ng/tumor (30.1 ± 9.6 , mean \pm SEM, n=3). Furthermore, because subcutaneous Neuro2a tumors are treated with intratumoral injections with 5×10^4 pfu of T-mfIL12, we newly measured the intratumoral IL-12 levels at this dose and found that the highest intratumoral IL-12 level was 0.668 ng/tumor (0.238 ± 0.147 , n=4). Therefore, in addition to the recombinant IL-12 dose used for former Supplementary Figure 5 (0.5 µg [500 ng]), we performed *in vivo* experiments using recombinant IL-12 at doses of 50ng and 1 ng. The results are now shown in new **Figure 6**, including the former Supplementary Figure 5 results as **Figure 6A**. Results were similar to 0.5 µg (500 ng) when the dose of 50 ng was used for rIL-12 (new **Figure 6B**). When the dose of 1 ng was used for rIL-12, a tendency similar to 500 ng and 50 ng was observed, except that rIL-12 alone showed no significant efficacy in both treated and untreated sides, and that T-mfIL12, but not T-01+rIL-12, was significantly more efficacious than rIL-12 alone in the treated side ($p < 0.001$, two-way ANOVA; new **Figure 6C**).

A new section is added to the Results as the last section (p.25, 2nd paragraph):

Comparison of in vivo efficacy of T-mfIL12 and intratumoral recombinant IL-12 administration

To evaluate the benefit of expressing IL-12 as an armed oncolytic HSV-1, we compared the efficacy of T-mfIL12 with that of direct intratumoral injections with recombinant murine IL-12 (rIL-12) in A/J mice harboring bilateral subcutaneous Neuro2a tumors (**Fig. 6**). **It has been reported using immunocompetent mice bearing poorly immunogenic subcutaneous tumors that 500 ng of rIL-12 can cause tumor growth inhibition when injected intratumorally³⁴. From the above**

experiment of Figure 5B, the highest intratumoral IL-12 level by intratumoral injection with 2×10^6 pfu of T-mfIL12 was calculated to be 45.7 ng/tumor (30.1 ± 9.6 , mean \pm SEM). Further, when subcutaneous Neuro2a tumors were treated with intratumoral injections with 5×10^4 pfu of T-mfIL12, the highest intratumoral IL-12 level was measured to be 0.668 ng/tumor (0.238 ± 0.147 , $n=4$). Hence, for rIL-12, we tested three different doses 500 ng, 50 ng and 1 ng. When established tumors reached approximately 5 mm in diameter, rIL-12, T-01 (5×10^4 pfu) without or with rIL-12, T-mfIL12 (5×10^4 pfu) or mock was inoculated into the left tumors only on days 0 and 4 ($n=10$ per group). When the dose of 500 ng was used for rIL-12, rIL-12, T-01, T-01+rIL-12 and T-mfIL12 were all significantly more efficacious than mock in the treated side ($p<0.001$ vs mock for all, two-way ANOVA; Fig. 6A). In the untreated side, rIL-12 alone showed no significant antitumor effect compared with mock, whereas T-01+rIL-12 and T-mfIL12 showed a significantly higher efficacy than mock ($p=0.039$ and $p<0.001$ vs mock, respectively, two-way ANOVA). Further, in the untreated side, T-mfIL12, but not T-01+rIL-12, was significantly more efficacious than rIL-12 alone ($p=0.003$, two-way ANOVA). Similar results were obtained when the dose of 50 ng was used for rIL-12 (Fig. 6B). When 1 ng was used for rIL-12, the dose representing the intratumoral IL-12 level treated with T-mfIL12 used in these experiments, rIL-12 alone showed no significant efficacy in both treated and untreated sides, and T-mfIL12, but not T-01+rIL-12, was significantly more efficacious than rIL-12 alone in the treated side ($p<0.001$, two-way ANOVA; Fig. 6C). These results implicate that virus-encoded IL-12 acts more effectively than recombinant IL-12 on both local and systemic antitumor immunity.

Comment 3:

The immunohistochemistry data (Figure 5A) should be quantified.

Response:

The immunohistochemistry in Figure 5A was performed to assess whether IL-12-expressing viruses function to promote infiltration of lymphocytes into the tumor, in response to this reviewer's question in the first revision (Major comment 9 of the first review). Qualitative results show that both IL-12-expressing viruses can indeed cause increased infiltration of CD4⁺ and CD8⁺ lymphocytes, more prominently with T-mfIL12 and T-mIL12-IRES than T-01. The immunohistochemistry was not performed to compare the strength of antitumor immunity induced by the two types of IL-12-expressing viruses: Because viruses were injected directly into tumors, quantification of immunopositive cells in certain sections would not be suitable for the purpose. Instead, we evaluated the strength of antitumor immunity induction by measuring the intratumoral IFN γ levels (Figure 5B) and by performing ELISpot assay using splenocytes (Figure 5C). Both

results show that T-mfIL12 induces significantly stronger antitumor immune responses than T-mIL12-IRES as already described in the Results.

To clarify the immunohistochemistry results, a sentence is modified in the Results (p.23, first paragraph):

An increased infiltration of CD4⁺ and CD8⁺ lymphocytes were observed in the tumor for all three viruses, both in the treated and the untreated side, **more prominently with T-mfIL12 and T-mIL12-IRES than T-01** (Fig. 5A).

Comment 4:

It seems the intratumoral levels of IL-12 are highest on the first day after treatment (Figure 5B). Is this from input virus? Non-purified virus suspensions can contain considerable amounts of transgene product.

This would not be transferable to an actual clinical treatment setting with highly purified/GMP-grade virus.

Response:

To address the reviewer's question, we measured the IL-12 concentrations in mock and virus suspensions of T-01, T-mfIL12, and T-mIL12-IRES. IL-12 was not detected in mock and T-01 (detection limit 7.8 pg/ml). The amounts of IL-12 in 20 μ l (2×10^6 pfu) of T-mfIL12 and T-mIL12-IRES were calculated to be 157.3 pg and 5.2 pg, respectively. On the other hand, from Figure 5B, the amounts of IL-12 in tumors treated with T-mfIL12 and T-mIL12-IRES on day 1 were 2590 ± 1450 pg/ml and 228 ± 115 pg/ml, respectively (mean \pm SD), which calculated to be 30100 ± 16600 pg and 770.5 ± 1130.6 pg per tumor, considerably higher (191-fold and 148-fold) than the amounts contained in virus suspensions. We further measured the IL-12 levels in the tumors on day 0, immediately after intratumoral virus injections, and the IL-12 levels were 7.7 ± 9.9 pg/ml for T-mfIL12 and not detected for T-mIL12-IRES. Therefore, the IL-12 detected on day 1 and after resulted from the IL-12 expression by the viruses.

Following sentences are added to the Methods, "*In vivo IL-12 and IFN γ levels*" section (p.15, 2nd paragraph):

Tumor samples were also collected for T-mfIL12 and T-mIL12-IRES groups on day 0, immediately after virus injections (n=4). The base IL-12 concentrations in mock and virus suspensions were measured by ELISA: IL-12 was not detected in mock and T-01, and the amounts of IL-12 in 20 μ l (2×10^6 pfu) of T-mfIL12 and T-mIL12-IRES were calculated to be 157.3 pg and 5.2 pg,

respectively.

Following sentences are modified or added to the Results (p.23, last paragraph):

In the tumor, the levels of IL-12 peaked on day 1, measuring 2590 ± 1450 pg/ml and 228 ± 115 pg/ml for T-mfIL12 and T-mIL12-IRES, respectively (mean \pm SD), and both IL-12 levels gradually decreased by days 3 and 6 (Fig. 5B). The intratumoral IL-12 levels on day 0, immediately after virus injections, were 7.7 ± 9.9 pg/ml for T-mfIL12 and not detected for T-mIL12-IRES, therefore the IL-12 detected on day 1 and after resulted from the IL-12 expression by the viruses.

Comment 5:

Figure 2D (in vitro) only includes one timepoint, no kinetics or the same timepoints as the in vivo data, so these data cannot be compared and it is unclear which levels of IL-12 are actually reached by virus replication.

The authors state that "The temperature was shifted to 39.5°C to prevent viral replication for ELISA sampling."

Then the ELISA cannot measure the levels that are reached during viral replication.

So the same question applies to the in vitro data: Is the IL-12 that was detected from input virus?

Response:

The purpose of the experiment shown in Figure 2D was to compare the amount of IL-12 expression by two types of IL-12 expressing virus in murine cell lines. The temperature 39.5°C was used exactly for this purpose of detecting the expression of IL-12 without the effect of virus replication. As we described in our response to this reviewer's question in the first revision (Minor comment 1), all viruses used in this study derive from temperature sensitive HSV-1 strain F and do not replicate at 39.5°C. It is shown in Figure 3 that there is no difference in cytopathic activities (reflecting no difference in replication capabilities) of these two types of viruses in the same murine cell lines. The Figure 2D study was performed at MOI=1 so that, theoretically, most cells are infected with the virus. Because the viruses are oncolytic, the majority of infected cells are killed within 48h at 37°C as shown in Figure 3. The temperature 39.5°C was used to allow cells to express IL-12 without being destroyed. The supernatant was collected at 48h to allow as much expression time as possible. For the purpose of this experiment, we consider it not worthwhile to perform additional experiments at multiple timepoints or at replication-competent temperature (37°C).

As described above in our response to Comment 4 of this reviewer, the amounts of IL-12 in virus suspensions used for these *in vitro* experiments were negligible, so the IL-12 detected in Figure 2D

resulted from the IL-12 expression by the viruses.

Comment 6:

Also, I would not use the term "conditioned media" but rather "supernatants" for these samples.

Response:

As the reviewer suggested, we changed the term "conditioned media" to "supernatants" throughout the manuscript.

Major point 7:

The authors have changed how they calculate tumor volume. Was this not pre-specified in the animal protocol?

They now state "Tumor growth was determined by measuring tumor volume (length × width × height × 0.52) twice a week" - why 0.52?

Response:

The animal protocol prespecifies that the animals are to be sacrificed when the maximum diameter of the tumor reached 22 mm, not the tumor volume. We merely changed how to express the tumor volume in Figures, so that the expressed tumor volumes are close to actual tumor volumes.

The equation for ellipsoid is

$$\frac{3}{4} \times \pi \times \frac{1}{2} \text{ length} \times \frac{1}{2} \text{ width} \times \frac{1}{2} \text{ height}$$

which is equal to

$$\text{length} \times \text{width} \times \text{height} \times 0.52$$

(π [Pi] is the circle ratio [approximately 3.14]).

Responses to Reviewer #2

Comment

The authors have satisfactorily answered all my concerns.

Response

We thank the reviewer for finding our revisions satisfactory.

Responses to Reviewer #3

Comment

The authors have taken time to take on board the comments and suggestions made by the reviewers. The manuscript has been significantly improved and most of the reviewers comments have been addressed with the inclusion of new data and necessary corrections. The revised version is worthy of publication.

Response

We thank the reviewer for the kind comments and for finding our revisions satisfactory.

REVIEWERS' COMMENTS:

Reviewer #1 (Remarks to the Author):

No further comments.

December 29, 2022

RE: COMMSMED-22-0076B Revised

Fusion peptide is superior to co-expressing subunits for arming oncolytic herpes virus with interleukin 12

by Hiroshi Fukuhara, Yuzuri Tsurumaki Sato, Jiangang Hou, Miwako Iwai and Tomoki Todo

Reviewer #1 (Remarks to the Author):

No further comments.

Responses to Reviewer #1

We thank the reviewer for re-reviewing our manuscript and for finding our revisions satisfactory.